# EffSelect: Efficient Feature Value Selection for Deep Recommender Systems with Mini-Batch Training

## Abstract

Features are critical to the performance of deep recommender systems, where they are typically represented as low-dimensional embeddings and fed into deep networks for prediction. However, a major challenge remains unaddressed: the sparsity and long-tail distribution in feature data result in a large number of non-informative feature values. These redundant values significantly increase memory usage and introduce noise, thereby impairing model performance. Most feature selection or pruning methods operate at a coarse granularity, either selecting entire features or fields, while finer-grained methods require a large number of additional learnable parameters. These methods struggle to effectively handle pervasive redundant features. To address these issues, we introduce **EffSelect**, a novel framework for finer-grained selection method at the level of feature values. Unlike previous methods, EffSelect directly quantifies the *contribution to the prediction loss* of each feature value as its importance. Specifically, we propose a mini-batch pre-training strategy that requires only 5% of the data for rapid warm-up, enabling real-time adaptation. Using the trained model, we introduce an efficient and robust gradient-based mechanism to evaluate feature value contribution, discarding those features with low scores. EffSelect is theoretically guaranteed and achieves superior performance without introducing any additional learnable parameters to the base model. Extensive experiments on benchmark datasets validate the efficiency and effectiveness of **EffSelect**. Code is available at `https://anonymous.4open.science/r/EffSelect_ICLR/`.

## 1 Introduction

Modeling the features of given data is crucial for practical recommendation tasks (Wang et al., 2025b; Wu et al., 2024; Du et al., 2024). With the development of deep models, researchers recognize the vast potential of deep models in capturing complex features and their interactions, leading to the design of advanced Deep Recommender Systems (DRSs) (Cheng et al., 2016; Guo et al., 2017). In these deep networks, features from each field (a feature column, e.g., "Gender" or "Age") are typically encoded and transformed into low-dimensional vectors before being fed into subsequent layers (Zhao et al., 2021; Zhaok et al., 2021). Many pioneers have focused on improving network architectures, such as the CrossNet paradigm proposed by DCN (Wang et al., 2017) and the integration of a feature weighting module in MaskNet (Wang et al., 2021). However, despite the extensive research on model architecture optimization, automatic feature-level optimization remains partially explored. One important issue is feature redundancy which can hamper the model's ability to learn interaction patterns and impact performance (Chen et al., 2016; Zhu et al., 2022; Wang et al., 2025a), as the redundancy kept in the embedding table (Jia et al., 2024; Wang et al., 2025c).

To reduce feature redundancy, feature selection methods are proposed and generally categorized into two types based on selection granularity. The coarse-granularity type is **feature field selection**, traditional methods (*e.g.*, XGBoost/RFE), and other field-level selection frameworks (Wang et al., 2022) inspired by Neural Architecture Search (NAS) typically remove entire redundant feature fields. These methods fail to distinguish the heterogeneous importance of distinct feature values (*e.g.*, "Male" and "Female") within the same field (*e.g.*, "Gender"), often leading to collateral selection errors. In such cases, critical features may be discarded alongside irrelevant ones, or vice versa.

In contrast, the fine-granularity Feature value selection methods (Liu et al., 2021) like OptFS (Lyu et al., 2023) go beyond field-level constraints. They assign trainable gates for each feature value and remove values with small weights after training. They are limited by initialization dependencies based on the *Lottery Ticket Hypothesis* (Malach et al., 2020) [1], which restricts retraining flexibility. Moreover, assigning independent learnable parameters to each feature or value increases computational overhead, conflicting with the dynamic nature of real-world recommender systems that require frequent data updates. More fundamentally, the joint optimization of gating mechanisms and embedding representations often results in competing training objectives and impairs the model's convergence because the gating regularization tends to learn small weights and create a bottleneck for embedding utilization.

In summary, we identify two main issues with existing feature selection methods: 1) **Low efficiency**. These methods require extensive pretraining, and the selection results heavily depend on pre-trained embeddings or gates, which do not meet the needs of recommender systems that require fast iteration and quick estimation of feature importance. 2) **Bad robustness**. Gate-based feature value selection methods are highly sensitive to hyperparameters, lacking robust performance guarantees. What's worse, the learning of gates and the updating of embedding tables are interdependent, which amplify gradient errors and excessively relying on the training set. An intuitive approach is to set the embedding corresponding to each feature value to zero or a random value, and observe the impact on the prediction or loss to assess their contribution one by one. However, the number of feature values in recommender systems can reach tens of millions, making this approach impractical. Therefore, it is necessary to directly obtain the actual contribution of each feature value.

To address the aforementioned challenges, we propose EffSelect, an efficient and effective feature value selection framework. To efficiently determine the importance of feature values, we select mini-batches that cover most features while preserving the feature distribution to pre-train the model and embedding table. This approach eliminates the need for gate-based methods or the stringent requirements of fine-grained learning typically associated with pre-training, allowing for obtaining feature importance with only a small amount of training data. Subsequently, we propose the FeatIS module, which provides a reference starting point for non-informative features and calculates the contribution of each feature value to the final loss based on the gradient. To obtain more accurate estimates, we extend FeatIS by using integral approximation to provide a more precise estimation of feature value contributions. All feature values are then sorted in descending order of importance, and only the top ones are selected. In summary, our contributions are as followed:

- **For efficiency**, we propose a batch selection scheme based on feature value coverage maximization. This approach ensures consistency with the original data distribution during sampling while achieving broad coverage of feature values, thereby enabling rapid pre-training of the embedding table and reduce the parameters of re-trained model.

- **For robustness**, we compute the loss term based on the fundamental theorem of calculus and map the contribution of the loss term to each feature value using a divide-and-approximate method. This approach is grounded in more solid theoretical foundations and requires only backpropagation on the validation dataset, without any additional learnable parameters.

- Extensive experiments on four benchmark datasets demonstrate the efficiency of our method in importance computation and its robustness in feature value estimation.

## 2 RELATED WORK

### 2.1 FEATURE SELECTION IN DRSS

[2] For effective feature selection, many works learn feature field importance through sensitivity or gates. Permutation Feature Importance (PFI) (Fisher et al., 2019) is a simple method based on performance sensitivity. It requires a well-trained model and then, for each batch, shuffles each field and observes the impact on prediction performance. This change (*e.g.*, $\Delta$AUC) is considered the importance of the field. With the rise of Neural Architecture Search (NAS) (Zoph & Le, 2022), researchers have attempted to simulate the process of field selection using search techniques. For

---

[1]Further details can be found in Appendix A.2.

[2]For more detail of the background on feature field and feature value, please refer to Appendix A.1.

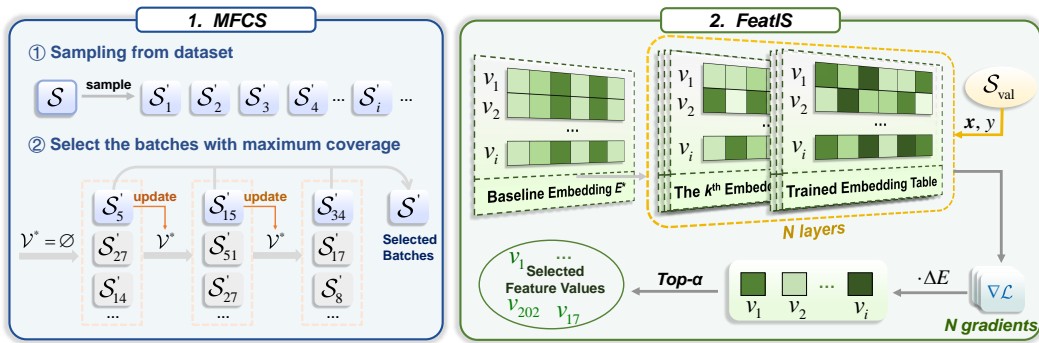

Figure 1: The main framework legend for EffSelect. EffSelect consists of two main parts: MFCS and FeatIS. On the left, mini-batches are selected for network and embedding warm-up, while on the right, the importance of different feature values is analyzed on the validation set.

example, AutoField (Wang et al., 2022) uses a graph with select or not nodes to learn field importance. Beyond it, many autoencoder-based feature selection methods are proposed to learn a gate or network to obtain the feature importance from different granularity (Balın et al., 2019; Nilsson et al., 2024; Chen et al., 2018). To achieve stronger adaptability, AdaFS (Lin et al., 2022) designs a controller that learns weights for each sample's fields, instead of sharing the same importance across all samples. MvFS (Lee et al., 2023) improves AdaFS by setting multiple controllers from different views. Despite improvements, these methods still operate at the feature field level. Existing works on feature value (Liu et al., 2021) optimization have some pioneering contributions, but their practical use remains limited. For example, OptFS (Lyu et al., 2023) designs learnable gates and functions for each feature value. It introduces a large number of additional learnable gates and hyperparameters to tune, making the application sensitive to prior settings. Therefore, the ideal method is one that minimizes the introduction of extra learnable parameters for efficient training, reduces hyperparameters to enhance robustness, and provides better guarantees in performance, which is the goal of the proposed method in this paper.

## 2.2 MINI-BATCH SELECTION

Mini-batch training uses a small fraction (*e.g.*, $\rho$) of the total dataset to train the model, speeding up the training process. The selected data, $\mathcal{S}'$, is a subset of the full dataset, $\mathcal{S}$, where the ratio of selected to total data is at most $\rho$ ($\frac{|\mathcal{S}'|}{|\mathcal{S}|} \leq \rho$). Previous works (Kirsch et al., 2019; Mirzasoleiman et al., 2020; Yang et al., 2024) on mini-batch training focus on achieving performance similar to that of full-batch training using a small amount of data. Given a data distribution $P$ and a loss function $\mathcal{L}$, the goal is to minimize the loss on the model $\Theta_{\mathcal{S}'}$ trained with the mini-batch $\mathcal{S}'$, as:

$$\min_{\mathcal{S}' \subset \mathcal{S}: \frac{|\mathcal{S}'|}{|\mathcal{S}|} \leq \rho} \mathbb{E}_{\boldsymbol{x}, y \sim P} \mathcal{L}(y, \hat{y}; \Theta_{\mathcal{S}'}), \tag{1}$$

where $\boldsymbol{x}$ represents the input features, $y$ represents the true labels, and $\hat{y}$ represents the predicted labels from the model.

However, previous approaches require **training on the complete dataset** followed by retraining on selected data subsets. In our paper, we propose a training-free selection method, with maximum feature value coverage on $\mathcal{S}'$, meaning that as many feature values as possible are trained during the process. The goal is to:

$$\max_{\mathcal{S}' \subset \mathcal{S}: \frac{|\mathcal{S}'|}{|\mathcal{S}|} \leq \rho} \left( \frac{|\mathcal{F}(\mathcal{S}')|}{|\mathcal{F}(\mathcal{S})|} \right), \tag{2}$$

where $\mathcal{F}(\cdot)$ means feature values for the given data. The iteration process terminates when $\frac{|\mathcal{S}'|}{|\mathcal{S}|} \leq \rho$.

## 3 FRAMEWORK

The logic of the EffSelect framework is to first train (or pre-warm) the embedding table and models using mini-batch data, and then analyze the impact of embedding table features, sorting those that activate larger gradients with respect to the prediction. In this section, we first describe how to train the base model in a mini-batch setting. Then is followed by how to obtain the feature value importance using the mini-batch data trained model.

### 3.1 MAXIMIZING FEATURE COVERAGE SAMPLING (MFCS)

To train the embedding table effectively with a few batches, we hold that mini batches should satisfy the following two basic properties:

**Proposition 1.** *Distribution Consistency. The samples forming a mini-batch should not distort the data distribution. The distribution $P_{\mathcal{S}'}$ of the mini-batch data set $\mathcal{S}'$ should match the distribution $P_{\mathcal{S}}$ of the entire data set $\mathcal{S}$, as $P_{\mathcal{S}'} = P_{\mathcal{S}}$. The consistency can be confirmed with $\rho$ suggested in Appendix C.2.*

**Proposition 2.** *Feature Coverage Maximum. As shown in Equation 2, with the dataset ratio $\rho$, the feature value coverage should be maximized.*

We design a greedy algorithm to achieve maximum feature value coverage. To comply with Proposition 1, We first perform a sampling without replacement on the total dataset $\mathcal{S}$ in batches, where the batch size is $B$. The total number of batches is $\left\lceil \frac{|\mathcal{S}|}{B} \right\rceil$, as $\mathcal{S}'_b \sim P_{\mathcal{S}}, b = 1, 2, \ldots, \left\lceil \frac{|\mathcal{S}|}{B} \right\rceil$. Since it is sampling without replacement, we have:

$$\mathcal{S}'_i \cap \mathcal{S}'_j = \emptyset, \quad \forall i \neq j. \tag{3}$$

Thus the data within these batches are guaranteed to maintain consistency with the original distribution. Based on this, to further achieve Proposition 2, we precompute the feature values in each batch and select the batch that contains the most feature values. Let the current selected feature values set be denoted as $\mathcal{V}^*$, with the initial condition $\mathcal{V}^* = \emptyset$. Then the selected batch index can be formulated as:

$$\max_b |\mathcal{F}(\mathcal{S}'_b) \cup \mathcal{V}^*|, b = 1, 2, \ldots, \left\lceil \frac{|\mathcal{S}|}{B} \right\rceil. \tag{4}$$

Based on the feature values in the $b$-th batch, we update the current feature values set as: $\mathcal{V}^* \leftarrow \mathcal{F}(\mathcal{S}'_b) \cup \mathcal{V}^*$. Then, according to the scheme in Equation 4, batches are iteratively selected that can bring the most additional features compared to the current feature values set. The final selected data samples form the union of all $\cup \mathcal{S}'_i$ (*a.k.a.* $\mathcal{S}'$). In fact, this indicates that $\mathcal{F}$ is a **submodular function**. The relevant proof and its theoretical upper bound are provided in Appendix C.

For MFCS, the process of selecting the batch containing the most feature values involves two **linear** steps: one for calculating the additional feature values of the current batch relative to $\mathcal{V}^*$, and the other for selecting the batch that yields the most additional feature values. The latter incurs minimal time cost, as the total number of batches is small, but the calculation of additional feature values often involves higher costs. Since the feature values of each batch remain unchanged during the selection process, and the preprocessed and encoded feature values are discretized, MFCS can be optimized using `bitmap`.

A `bitmap` is a $\{0, 1\}^N$, where $N$ represents its length, and the 0/1 at each position indicates the presence or absence of the corresponding feature value. For each feature field in every batch, we use a bitmap of length equal to the maximum feature value index to represent which features are included in the current batch. Additionally, the features in the currently selected batch, $\mathcal{V}^*$, are also maintained using a `bitmap`. This way, when evaluating the number of additional feature values, there is no need to use a set for maintenance. The whole step of MFCS is shown in Algorithm 1.

The iteration process terminates when $\frac{|\mathcal{S}'|}{|\mathcal{S}|} > \rho$. With the selected batch data $S'$, the base model (*e.g.*, DCN, MaskNet) is trained using the cross-entropy loss function. This process aims to train the model parameters and the embedding table as much as possible, preparing for the next step of the feature value scoring process in the embedding latent space.

$$\min_{\Theta_{\mathcal{S}'}} \frac{1}{|\mathcal{S}'|} \sum_{\boldsymbol{x},y \in \mathcal{S}'} \mathcal{L}_{\text{CE}}(y, \hat{y}; \Theta_{\mathcal{S}'}), \tag{5}$$

where $\mathcal{L}_{\text{CE}}$ is:

$$\mathcal{L}_{\text{CE}}(y, \hat{y}; \Theta_{\mathcal{S}'}) = - \left[ y \log(\hat{y}) + (1 - y) \log(1 - \hat{y}) \right], \tag{6}$$

and the $\hat{y}$ is predicted by the model parameterized with $\Theta_{\mathcal{S}'}$.

### 3.2 FEATURE VALUE IMPORTANCE SCORER (FEATIS)

#### 3.2.1 IMPORTANCE DESIGN

With the **well-trained** $\Theta_{\mathcal{S}'}$ including the embedding table, we expect to measure the contribution of each feature value. An intuitive approach is to sequentially mask each feature value's embedding while keeping others unchanged, then measure the resulting validation loss difference. For a feature value $v \in \mathcal{V}$, this can be formulated as:

$$\Delta \mathcal{L}(E) = \mathcal{L}(E - \mathbb{I}_v \odot E) - \mathcal{L}(E), \tag{7}$$

where we let $\mathcal{L}$ (indeed is $\mathcal{L}_{\mathcal{S}_{\text{val}}}$ on the validation dataset) **is the function of embedding table** $E$ here[3], $E$ is formed by concatenating $E_v$ for $v \in \mathcal{V}^*$, and $\mathbb{I}_v$ is an indicator vector (or a vector consisting of 1's and 0's). This operation eventually sets the embedding corresponding to the feature value $v$ to 0, while keeping the other positions unchanged. Though feasible, it is impractical to calculate the importance of each feature value individually for large datasets with hundreds of thousands or even millions of feature values.

If the loss term is viewed as a multivariate function of each feature value, the contribution of each value can be measured using a Taylor expansion. Mathematically, from the perspective of Taylor expansion, the contribution of the embedding $E_v$ of each feature value $v$ to the loss can be expressed as:

$$\mathcal{L}(E) = \underbrace{\mathcal{L}(E^*)}_{\text{identical}} + \sum_v \underbrace{\nabla_{E_v} \mathcal{L}(E^*) \cdot (E_v - E_v^*)}_{\text{different}} + \mathcal{O}(|E_v - E_v^*|^2) \tag{8}$$

where $E_v^*$ is the starting point embedding for the feature value $v$, and $E^*$ is concatenated by each $E_v^*$. In the Taylor expansion, it is the starting point of the expansion. From Equation 8, we could find that for each feature value $v$, the term $\mathcal{L}(E^*)$ is **identical**, therefore, the 1-st term of Taylor Expansion essentially describes the contribution of feature value $v$:

$$I_v = \left| \nabla_{E_v} \mathcal{L}(E^*) \cdot (E_v - E_v^*) \right|. \tag{9}$$

However, the limitation of this measurement lies in its neglect of higher-order terms in Equation 8 with respect to the feature value $v$. These higher-order terms are computationally unfriendly, as their consideration would involve higher-order joint gradients between feature values $v_i$ and $v_j$, which would escalate the time complexity from linear (for a single gradient backpropagation) to polynomial. In practice, this approach is infeasible due to the typically large size of $|\mathcal{V}^*|$ in real-world data. Therefore, a more precise measurement is needed to attribute contributions to individual feature values, ensuring that the contribution of each feature value can be computed within a single gradient backpropagation, and the error upper bound is both **controllable** and **tractable**.

Revisiting the $\Delta \mathcal{L}$ term in Equation 7, if we interpret the function $\mathcal{L}$ as the antiderivative in calculus, and consider the impact of each feature value $v$ on the loss, then according to the *Newton-Leibniz formula*, we know that:

$$\mathcal{L}(E) - \mathcal{L}(E^*) = \int_{E^*}^{E} \nabla_X \mathcal{L}(X) \cdot dX. \tag{10}$$

Based on this, we only need to compute the value on the right-hand side of the equation and attribute it to each feature value $v$. Inspired by Sundararajan et al. (2017), we innovatively adopt numerical integration (Morokoff & Caflisch, 1995) to the right-hand side, as:

---

[3]From this section of importance calculation, the loss function $\mathcal{L}$ is calculated on the **validation dataset**, which is different from the previous training stage.

$$\int_{E^*}^{E} \nabla_X \mathcal{L}(X) \cdot dX \approx \sum_{k=0}^{N-1} \nabla_X \mathcal{L}(X_{t_k}) \cdot \left(X_{t_{k+1}} - X_{t_k}\right), \tag{11}$$

where $X_{t_k}^{(m)}$ is the value of the $m$-th path at discrete point $t_k$, $\nabla_X \mathcal{L}(X_{t_k}^{(m)})$ is the gradient of the loss function $\mathcal{L}$ at $X_{t_k}^{(m)}$, $M$ is the total number of random paths, and $N$ is the number of discrete points on each path.

In this way, we can transfer the Taylor expansion error with the integration error. However, directly solving using Equation 11 is computationally expensive. To simplify the process, we choose the linear path from $E^*$ to $E$, and compute the original integral using a *divide-and-approximate* method, which makes the loss term become:

$$\mathcal{L}(E) - \mathcal{L}(E^*) \approx \sum_{k=1}^{N} \nabla_X \mathcal{L}\left(E^* + \frac{k}{N}(E - E^*)\right) \cdot \frac{E - E^*}{N}. \tag{12}$$

For the importance of each feature value, we take the corresponding term for $v$ in the above equation. The final importance is defined as:

$$I_v = \sum_{k=1}^{N} \left| \nabla_{E_v} \mathcal{L}\left(E^* + \frac{k}{N}(E - E^*)\right) \cdot \frac{E_v - E_v^*}{N} \right|. \tag{13}$$

where related symbols have been explained in Equation 11. Together with Equation 9, these form two variants of our method. Specifically, when $N = 1$, Equation 13 degrades to Equation 9. In Appendix B.1, we formally prove why it can maintain a lower approximation error and theoretically demonstrate why this approach may perform better compared to Equation 7.

### 3.2.2 THE CHOICE OF $E^*$

As discussed in previous subsections, the role of $E^*$ is to serve as a "starting point" for measuring the importance of feature values. For each feature value $v$, there is an $E_v^*$ in $E^*$. It should contain the least information to highlight the importance of each feature value. A simple approach is to choose an embedding like zerolike$(E_v)$ as $E_v^*$.

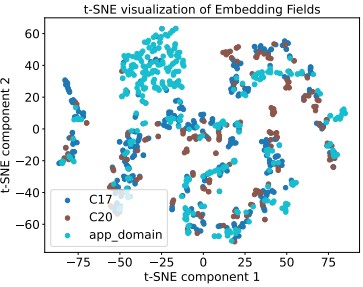

Figure 2: Embeddings t-SNE of feature values after pre-training with Avazu on Wide & Deep.

However, a potential issue is that the global zerolike $E^*$ may not necessarily contain the least information, as it could still influence predictions (*e.g.*, towards positive or negative records). As shown in Figure 2, the embeddings for each $v$ after pre-training exhibit distribution differences. The distributions of C17 and C20 are similar, while app_domain is more widely spread compared to the previous two.

Based on this observation, we propose using the field-wise mean value as the non-informative embedding for each feature $v$. According to the principle of maximum entropy, the most reasonable probability distribution under known constraints is the one with the highest entropy. The mean value, being the first moment, represents the central location of the data. In the absence of additional information, using the mean as a reference can be viewed as a "zero-order approximation," assuming that the data are symmetrically distributed around the mean, which aligns with the "unbiased assumption" in the maximum entropy principle.

Specifically, the field-wise mean value is the average embedding of all values corresponding to a feature field. Formally, we let $F_v$ be all feature values belonging to the same feature field, then $E_v^*$ can be represented as:

$$E_v^* = \frac{1}{|F_v|} \sum_{v_i \in F_v} E_{v_i}, \tag{14}$$

which means that the starting point embeddings $E_v^*$ of values within the same feature field are identical.

### 3.2.3 SORTING WITH $I_v$

Based on the obtained scores $I_v$ for each feature value $v$, we combine all feature values from different fields, sort them in descending order of their scores, and select the top for model learning. This approach is justified by the fact that the importance we design is directly related to the final loss, which ensures that the importance across different fields is on the same scale.

### 3.3 ALGORITHM COMPLEXITY ANALYSIS

The time complexity of EffSelect consists of two main parts. Let $l$ represent the total number of feature values in the complete training set. Since a `bitmap` is used, an $l$-length vector can mark the presence or absence of each feature in a mini-batch. In the MFCS phase, a total of $\rho \left\lceil \frac{|\mathcal{S}|}{B} \right\rceil$ batches need to be selected. For each selection, the contribution of all batches is computed, and then sorted to pick the one with the highest contribution. Therefore, the total complexity is $\rho \left\lceil \frac{|\mathcal{S}|}{B} \right\rceil^2 l$. In the importance score calculation phase, $\rho$ fraction of samples are used for training, and gradient backpropagation is performed $N$ times on the validation set. The time complexity is $\rho T_{\text{train}} + N T_{\text{val}}$. The total time complexity is the sum of these two parts. Since the computation of the $N$ segments can be parallelized, the actual process can be also optimized.

## 4 EXPERIMENT

### 4.1 SETTING UP

Table 1: Statistics of four datasets used for evaluation.

| Dataset | #Fields | #Training | #Validation | #Test | Positive% |
|---------|---------|-----------|-------------|-------|-----------|
| iPinYou | 16 | 13,195,935 | 2,199,323 | 4,100,716 | 0.08% |
| Ali-CCP | 23 | 42,299,905 | 21,508,307 | 21,508,307 | 3.89% |
| Avazu | 24 | 32,343,172 | 4,042,897 | 4,042,898 | 16.98% |
| Criteo | 39 | 36,672,493 | 4,584,062 | 4,584,062 | 25.62% |

For feature field selection methods, due to their inability to perform feature selection at the fine-grained level like feature value selection methods, we adopt different approaches. To ensure a fair comparison with baselines, for methods such as RF, XGBoost, RFE, and PFI, we select the feature fields corresponding to the point where the cumulative feature importance first exceeds 10% based on the feature field importance ranking. In the case of AdaFS and MvFS, we retain the most important **10%** of features for each sample. For feature value selection methods, we select the top 10% of the most important feature values for training and evaluation. The hyperparameter config can be found in Appendix D.3, and the detailed introduction of these methods can be found at Appendix D.2. We evaluate the effectiveness of the proposed methods using two classic base models in real recommender system scenarios: DCN (Wang et al., 2017) and MaskNet (Wang et al., 2021). Due to the space limitation, the comparision with autoencoder-based feature selection methods is shown in Appendix E.1.

### 4.1.1 DATASET

As shown in Table 1, we select four benchmark datasets to evaluate the effectiveness of EffSelect. They are iPinyou, Ali-CCP, Avazu, and Criteo. The brief situation of the datasets is shown in the table, and the details can be found in Appendix D.1. Note that since the iPinYou dataset only provides the training and test sets by default, to ensure the reliability of the results, we additionally split 1/7 of the training data as a validation set. For all datasets, the low-frequency filter threshold is set to 2.

### 4.2 MAIN RESULTS

In this section, we examine the impact of different feature selection methods on the results under the condition of 10% features or fields. This setting is significant for inference on small edge devices and helps evaluate the effectiveness of feature selection methods when resources are extremely limited.

As shown in Table 2, EffSelect achieves the best performance in most cases. Although the difference from the Base Model results is relatively large on the Avazu dataset, it outperforms the baseline on the other three datasets. Specifically, traditional feature field selection methods struggle to select truly

Table 2: Comparison with different feature field and feature value selection method.

| Model | Dataset | Metrics | Base | Field Selection | | | | | | Value Selection | | |
|---|---|---|---|---|---|---|---|---|---|---|---|---|
| | | | | RF | XGBoost | RFE | PFI | AdaFS | MvFS | OptFS | EffSelect$_{\mathcal{Z}}$ | EffSelect$_{\mathcal{M}}$ |
| DCN | Criteo | AUC | 0.8090 | 0.7879 | 0.7793 | 0.8058 | 0.8034 | 0.7998 | 0.7996 | 0.8077 | **0.8102** | **0.8102** |
| | | Logloss | 0.4427 | 0.4610 | 0.4674 | 0.4457 | 0.4478 | 0.4514 | 0.4525 | 0.4439 | 0.4418 | **0.4417** |
| | Avazu | AUC | **0.7908** | 0.7076 | 0.7500 | 0.7717 | 0.7634 | 0.7823 | 0.7836 | 0.7877 | 0.7744 | 0.7737 |
| | | Logloss | **0.3735** | 0.4134 | 0.3953 | 0.3843 | 0.3880 | 0.3829 | 0.3830 | 0.3760 | 0.3829 | 0.3829 |
| | iPinYou | AUC | 0.7642 | 0.7383 | 0.7635 | 0.7319 | 0.7572 | 0.7391 | 0.7270 | 0.7624 | 0.7683 | **0.7699** |
| | | Logloss | 0.5630 | 0.5766 | 0.5656 | 0.5894 | 0.5662 | 0.5821 | 0.5988 | 0.5623 | 0.5620 | **0.5607** |
| | Ali-CCP | AUC | 0.5956 | 0.5762 | 0.5834 | 0.5743 | 0.5939 | 0.6004 | 0.6009 | 0.5979 | 0.6000 | **0.6021** |
| | | Logloss | 0.1639 | 0.1631 | 0.1630 | 0.1640 | 0.1631 | 0.1656 | 0.1656 | 0.1644 | 0.1622 | **0.1621** |
| MaskNet | Criteo | AUC | 0.8098 | 0.7880 | 0.7722 | 0.8062 | 0.8009 | 0.7999 | 0.7999 | 0.8086 | 0.8110 | **0.8111** |
| | | Logloss | 0.4420 | 0.4609 | 0.4728 | 0.4453 | 0.4501 | 0.4509 | 0.4511 | 0.4431 | 0.4408 | **0.4407** |
| | Avazu | AUC | **0.7914** | 0.7129 | 0.7506 | 0.7724 | 0.7646 | 0.7834 | 0.7849 | 0.7900 | 0.7757 | 0.7766 |
| | | Logloss | **0.3731** | 0.4390 | 0.3950 | 0.3848 | 0.3876 | 0.3816 | 0.3808 | 0.3741 | 0.3824 | 0.3816 |
| | iPinYou | AUC | 0.7674 | 0.7242 | 0.7666 | 0.7534 | 0.7563 | 0.7580 | 0.7653 | 0.7570 | 0.7683 | **0.7699** |
| | | Logloss | 0.5608 | 0.5726 | 0.5628 | 0.5624 | 0.5646 | 0.5684 | 0.5629 | 0.5622 | 0.5598 | **0.5581** |
| | Ali-CCP | AUC | 0.6056 | 0.5739 | 0.5815 | 0.5733 | 0.5986 | 0.6020 | 0.5992 | 0.6005 | 0.6010 | **0.6109** |
| | | Logloss | 0.1637 | 0.1636 | 0.1630 | 0.1641 | 0.1660 | 0.1651 | 0.1661 | 0.1650 | 0.1624 | **0.1619** |

Since the loss on the iPinYou dataset is small, we use Logloss% instead of Logloss here. Base means using all feature values to train the model, EffSelect$_{\mathcal{Z}}$ means using zero-like starting point embedding to get the feature value importance, and EffSelect$_{\mathcal{M}}$ means using field-wise mean value as the starting point. The best results are in **bold** and the second is underlined.

useful subsets of feature values. This is understandable, as embedding tables gained popularity with the rise of deep learning, and earlier methods like XGBoost could only perform selection field-wise, without accounting for the contribution of different feature fields. AdaFS and MvFS yield relatively strong results, but these may largely depend on the model parameters from the pre-training stage, which contrasts with our method that independently retrains the model. Additionally, OptFS achieves relatively good performance with a masking mechanism, but its effectiveness is highly dependent on hyperparameter tuning. Overall, EffSelect achieves state-of-the-art performance in most cases.

## 4.3 EFFICIENCY

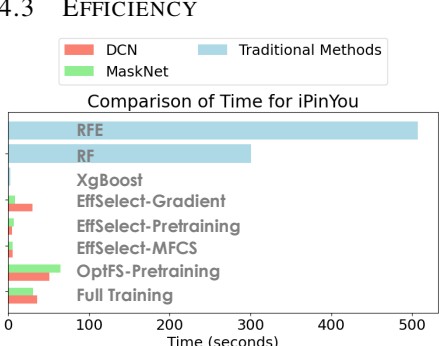

Figure 3: Time consumption. For traditional methods, it shows the total time.

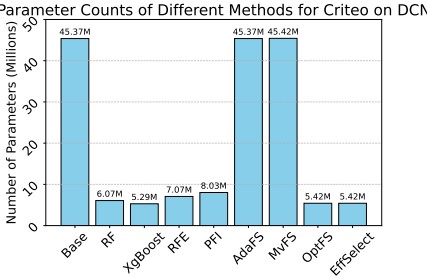

Figure 4: Parameter counts for different methods.

The advantage of EffSelect is clear: it can estimate feature importance using only a small number of batches. Meanwhile, both the time cost of each stage and the **parameter count** during re-training are also important. In this paper, we compute the **total parameter count**, since even zero-masked networks still participate in training structurally.

In Figure 3, there are significant differences in time cost among various methods. For other methods, it shows the time consumption per epoch. XGBoost is the fastest, while RF and RFE are much slower, though none of these achieve optimal performance. For EffSelect, on the iPinYou dataset, it uses very little time for batch selection and achieves much faster pre-training compared to full training. This shows a clear advantage over the gate-based approach used by OptFS.

As for parameter count (Figure 4), our method uses only about 12% of the original model's memory on the Criteo dataset with DCN. For XGBoost, its retraining footprint is slightly smaller than EffSelect, but its prediction performance is much lower. The method most similar to ours is OptFS, which is a strong baseline. It achieves good performance with the same parameter count as ours, although still worse. In contrast, AdaFS and MvFS generally require more parameters, as both rely on additional controllers that increase memory use and may slow down training.

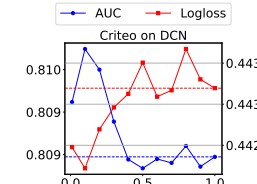 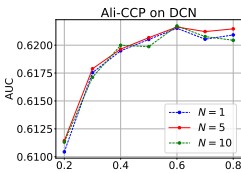 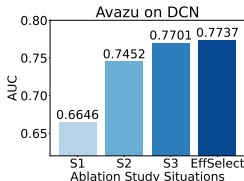 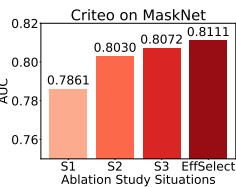

Figure 5: Ratio $\alpha$ influence with Criteo on DCN.

Figure 6: Ratio $\alpha$ influence with Ali-CCP on DCN with various $N$.

Figure 7: Ablation Study with Avazu on DCN.

Figure 8: Ablation Study with Criteo on MaskNet.

### 4.4 HYPERPARAMETER ANALYSIS

EffSelect involves three main hyperparameters: the proportion of the pretraining training set, $\rho$, the number of discrete points, $N$, and the ratio of selected feature values, $\alpha$. In each part of the study, we fix the settings of the other two. Unless otherwise specified, the default values are $\rho = 0.05$, $\alpha = 0.1$, and $N = 5$. The impact of $\alpha$ on the results is the largest. The dashed line in Figure 5 represents the AUC and Logloss using all feature values. On Criteo, using approximately 10% of the feature values achieves better prediction results than using all features. On Ali-CCP, with 60% feature values can bring 4.37% relative AUC improvement. This demonstrates great redundancy in the embedding table. However, the trend of overall performance trend varies significantly across different datasets. On Criteo, performance peaks at small $\alpha$ values and then shows a general downward trend. In contrast, on Ali-CCP, performance gradually increases and only shows significant degradation at large $\alpha$ values. This indicates that different datasets have different noise characteristics. Moreover, Figure 6 also shows that, as $\alpha$ changes, the segmentation number $N = 5$ generally performs better than $N = 1$. While $N = 10$ may have a minor advantage, the additional overhead makes it not worthwhile.

Since the impact of the other hyperparameter ($\rho$) on the results is smaller than that of $\alpha$, we have put it to Appendix E.2 to save space.

### 4.5 ABLATION STUDY

The ablation study of EffSelect consists of three main parts. **S1**: We randomly select 10% of the feature values for training. **S2**: We use the 5% selected by MCFS for pre-training, and the results are directly used as the final output. **S3**: We directly use the backpropagated gradients without multiplying by the change in embedding $E$ compared to $E^*$. These three parts evaluate the contribution of each component to the final result. EffSelect also uses 10% feature values in this experiment.

The influence of these three components on the final performance is evident. **S1** randomly selects 10% of the feature values and yields the worst performance, even when re-training with the full set of batches. This highlights the overall importance of EffSelect. For **S2**, using only the selected mini-batches is insufficient for the model to capture complex user history interactions. These mini-batches merely enable fast pre-training of the embedding layer. Optimal performance is achieved only when feature value selection is conducted on top of this and followed by full-data re-training. **S3** adopts an alternative strategy to measure feature importance, but its performance falls short of EffSelect. This is because it does not take loss sensitivity into account. Similar performance trends are observed across both datasets and both models.

## 5 CONCLUSION

Selecting a critical subset of feature values is essential for the performance and resource efficiency of recommender systems. Existing feature field and value selection methods either have coarse granularity or rely on gating mechanisms with low learning efficiency and robustness. To address these issues, we propose EffSelect, a framework that trains with mini-batches and uses the contribution of feature values to the loss function as a measure of feature value importance. This approach provides an efficient means for identifying and removing non-informative feature values. Experiments were conducted on four benchmark datasets using two base models, demonstrating that our method achieves optimal prediction performance in most cases. In addition, efficiency tests in terms of time and memory highlight the practical deployment advantages of EffSelect. Our work offers insightful ideas for selecting informative feature values with solid theoretical guarantee.

ETHICS STATEMENT

We confirm that our work adheres to the ICLR Code of Ethics [4]. Our study does not involve human subjects, nor does it raise any concerns related to privacy, security, or discrimination. The dataset used in this research is publicly available and has been properly credited. We have ensured compliance with all relevant legal and ethical guidelines, and there are no conflicts of interest related to sponsorship or affiliations. Our findings are presented with integrity, with all results accurately reported. If any ethical concerns arise during the review process, we are open to further discussion and clarification.

REPRODUCIBILITY STATEMENT

We are committed to ensuring the reproducibility of our work. All datasets used in the experiments are publicly available, and the data processing steps are described in detail. For the models and algorithms introduced, we provide a link to the source code [5], which is the same link provided in the Abstract.

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

## A BACKGROUND

### A.1 FEATURE FIELD AND FEATURE VALUE

In DRSs, feature selection consists of two parts: **feature field selection** and **feature value selection**. The main difference between them lies in the granularity of selection.

The former treats a feature field column as the unit of selection, where all feature values within the same field are either selected or dropped together. For example, the field *occupation* contains feature values like *teacher*, *doctor*, etc. Regardless of their individual contributions to model prediction, they are treated identically. On the other hand, feature value selection focuses on a finer granularity, considering each value of *occupation* individually for selection. Dropped feature values and any new feature values observed in validation or test sets are treated as `[OOV]` word and projected into an embedding.

### A.2 LOTTERY TICKET HYPOTHESIS

The lottery ticket hypothesis, proposed by Frankle and Carbin in 2018, states that: A randomly-initialized, dense neural network contains a subnetwork that is initialized such that — when trained in isolation — it can match the test accuracy of the original network after training for at most the same number of iterations.

According to (Malach et al., 2020), in mathematical terms, let $\mathcal{N}$ be a randomly-initialized neural network with weights $W$. There exists a subnetwork $\mathcal{N}_s$ of $\mathcal{N}$ with a subset of weights $W_s \subseteq W$ such that if we train $\mathcal{N}_s$ independently, the test accuracy of $\mathcal{N}_s$, denoted as $Acc(\mathcal{N}_s)$, is comparable to the test accuracy of $\mathcal{N}$, denoted as $Acc(\mathcal{N})$, after at most the same number of training iterations. That is:

$$Acc(\mathcal{N}_s) \approx Acc(\mathcal{N})$$

where the approximation is in terms of the performance on a given test dataset.

This hypothesis has significant implications. If true, it suggests that the process of training large neural networks can be made more efficient. Instead of training an entire large network, one could potentially find a good small subnetwork within it and then train only that subnetwork. However, finding such a "winning-ticket" subnetwork is **non-trivial**. Our proposed EffSelect does not rely on such assumptions or the tedious task of finding a subnetwork as kept feature values. It only requires 5% of the training data to distinguish informative features.

## B PROOF FOR FEATIS CONVERGENCE AND ERROR UPPER BOUND

The convergence and convergence rate of a method are crucial for its performance. The numerical integration method used in this paper can actually be implemented using the right endpoint rule or the composite midpoint rule. These two methods differ in convergence rates, and the former already yields good results in practice. Below, we will conduct an error analysis for both approaches.

### B.1 ERROR ANALYSIS OF THE RIGHT ENDPOINT RULE

Given a loss function $\mathcal{L}(E)$ with an embedding table $E = (E_v)_{v \in \mathcal{V}}$, we compare two estimators. The first is the **Taylor expansion** at the starting point $E^*$, which is expressed as:

$$\mathcal{L}(E) = \mathcal{L}(E^*) + \sum_v \nabla_{E_v} \mathcal{L}(E^*) \cdot (E_v - E_v^*) + \mathcal{O}(\|E - E^*\|^2), \tag{15}$$

where the first-order term accounts for the gradient of the loss function evaluated at $E^*$, and the second term represents the error of the approximation, which is of order $\mathcal{O}(\|E - E^*\|^2)$.

The second method we adopt as in Equation 13, the numerical integration with $N$ segments. The integral for the gradient $I_v$ is given by:

$$I_v = \sum_{k=1}^{N} \left| \nabla_{E_v} \mathcal{L} \left( E^* + \frac{k}{N} \Delta E \right) \cdot \frac{\Delta E_v}{N} \right|, \quad \Delta E := E - E^*. \tag{16}$$

Next, we aim to prove that the error upper bound of $I_v$ is smaller. We begin by assuming that the gradient of the loss function is Lipschitz continuous with respect to each specific feature value embedding. This assumption is reasonable because, during training, the computation from the embedding vector $v$ to the loss $L$ typically involves a sequence of continuous and differentiable operations (*e.g.*, matrix multiplication, addition, ReLU, Sigmoid, Softmax, etc.). Although ReLU is not differentiable at zero, it is piecewise linear, and its gradient is practically tractable. Therefore, assume that the gradient $\nabla \mathcal{L}$ is **Lipschitz continuous** with constant $L$. That is, for any two embedding vectors $E_1$ and $E_2$, we have:

$$\|\nabla\mathcal{L}(E_1) - \nabla\mathcal{L}(E_2)\| \leq L \cdot \|E_1 - E_2\|. \tag{17}$$

Then, consider the interval from $E^* + t_{k-1}\Delta E$ to $E^* + t_k\Delta E$, where $t_k = \frac{k}{N}$ and the step size is $\Delta t = \frac{1}{N}$. The difference between the two points is:

$$\|(E^* + t_k\Delta E) - (E^* + t_{k-1}\Delta E)\| = \|\Delta E\| \cdot (t_k - t_{k-1}) = \frac{1}{N}\|\Delta E\|. \tag{18}$$

According to the Lipschitz condition, the change in gradient is bounded by:

$$\|\nabla\mathcal{L}(E^* + t_k\Delta E) - \nabla\mathcal{L}(E^* + t_{k-1}\Delta E)\| \leq L \cdot \frac{1}{N}\|\Delta E\|. \tag{19}$$

Numerical integration approximates the integral in each interval using the gradient at the right endpoint. The approximation error arises from the variation of the gradient within the interval. For the $k$-th interval, the error term is:

$$\left| [\nabla\mathcal{L}(E^* + t_k\Delta E) - \nabla\mathcal{L}(E^* + t_{k-1}\Delta E)] \cdot \frac{\Delta E}{N} \right|. \tag{20}$$

By the **Cauchy–Schwarz** inequality, the absolute value of the dot product is bounded by the product of their norms:

$$\|\nabla\mathcal{L}(E^* + t_k\Delta E) - \nabla\mathcal{L}(E^* + t_{k-1}\Delta E)\| \cdot \frac{\|\Delta E\|}{N}. \tag{21}$$

Returning to Equation 20, the error upper bound is given by

$$L \cdot \frac{\|\Delta E\|}{N} \cdot \frac{\|\Delta E\|}{N} = \frac{L\|\Delta E\|^2}{N^2}. \tag{22}$$

The error in each subinterval is on the order of $\mathcal{O}\left(\frac{\|\Delta E\|^2}{N^2}\right)$, and there are $N$ subintervals in total. Therefore, the total error is:

$$\text{Total Error} \leq N \cdot \frac{L\|\Delta E\|^2}{N^2} = \frac{L\|\Delta E\|^2}{N}. \tag{23}$$

This shows that the error of the numerical integration is $\mathcal{O}\left(\frac{\|\Delta E\|^2}{N}\right)$. This shows that, in our application, we can theoretically reduce the error bound by increasing $N$, leading to more accurate estimates of feature value importance.

## B.2 ERROR ANALYSIS OF THE COMPOSITE MIDPOINT RULE

In practice, with the same number of segments $N$, the computational complexity remains the same. However, using the midpoint rule for integration gives a smaller error bound. We will prove that for a sufficiently smooth function $\mathcal{L}$, the error bound for the composite midpoint rule is:

$$\left| \mathcal{L}(E) - \mathcal{L}(E^*) - I^{\text{Mid}} \right| \leq \frac{K\|\Delta E\|^3}{24N^2} = \mathcal{O}\left(\frac{1}{N^2}\right) \tag{24}$$

where the constant $K$ originates from the bound on the third derivative of the function $\mathcal{L}$.

This requires a stronger assumption as followed. We need to assume that the function $\mathcal{L}$ is continuously differentiable enough and that its third-order derivative tensor is bounded. That is, there exists a constant $K > 0$ such that for all $X$:

$$\left\| \nabla^3 \mathcal{L}(X) \right\| \le K.$$

This assumption ensures that the rate of change of the Hessian matrix is bounded.

To prove this, first, we convert to a one-dimensional integral. We parameterize the path as $X(s) = E^* + s\Delta E$, where $s \in [0, 1]$, and convert the line integral into a standard one-dimensional integral. Let the step size be $h = 1/N$, then

$$\mathcal{L}(E) - \mathcal{L}(E^*) = \int_0^1 \nabla \mathcal{L}(X(s)) \cdot \Delta E \, ds. \tag{25}$$

We define the scalar function $g(s) := \nabla \mathcal{L}(X(s)) \cdot \Delta E$. The problem is thus transformed into calculating the approximation error for $\int_0^1 g(s) ds$. Then, consider the $k$-th interval $[s_{k-1}, s_k]$, which has a width of $h$ and a midpoint of $s^{(k)}$. The local error $\epsilon_k$ on this interval is:

$$\epsilon_k = \int_{s_{k-1}}^{s_k} g(s) \, ds - h \cdot g(s^{(k)}). \tag{26}$$

We expand $g(s)$ in a Taylor series around the midpoint $s^{(k)}$ (using the Lagrange remainder form):

$$g(s) = g(s^{(k)}) + g'(s^{(k)})(s - s^{(k)}) + \frac{g''(\xi_k)}{2}(s - s^{(k)})^2, \tag{27}$$

where $\xi_k$ is a point between $s$ and $s^{(k)}$. Substituting this expansion into the expression for $\epsilon_k$ and integrating, the linear term vanishes due to symmetry, leaving only the integral of the remainder term:

$$\epsilon_k = \int_{s_{k-1}}^{s_k} \frac{g''(\xi_k)}{2}(s - s^{(k)})^2 \, ds. \tag{28}$$

According to the Mean Value Theorem for Integrals, because $(s - s^{(k)})^2 \ge 0$, there exists an $\eta_k \in [s_{k-1}, s_k]$ such that:

$$\epsilon_k = \frac{g''(\eta_k)}{2} \int_{s_{k-1}}^{s_k} (s - s^{(k)})^2 \, ds, \tag{29}$$

and the integral of the quadratic term yields:

$$\int_{s_{k-1}}^{s_k} (s - s^{(k)})^2 \, ds = \frac{h^3}{12}. \tag{30}$$

Therefore, the local error is:

$$\epsilon_k = \frac{g''(\eta_k)}{2} \cdot \frac{h^3}{12} = \frac{g''(\eta_k) h^3}{24}. \tag{31}$$

The total error is the sum of all local errors. We take its absolute upper bound and let $G_{\max}$ be the maximum value of $|g''(s)|$ on $[0, 1]$, as:

$$\text{Total Error} = \left| \sum_{k=1}^N \epsilon_k \right| \le \sum_{k=1}^N |\epsilon_k| = \sum_{k=1}^N \left| \frac{g''(\eta_k) h^3}{24} \right| \le \sum_{k=1}^N \frac{G_{\max} h^3}{24} = N \cdot \frac{G_{\max} h^3}{24}. \tag{32}$$

Substituting $h = 1/N$, we could get:

$$\text{Total Error} \le N \cdot \frac{G_{\max}(1/N)^3}{24} = \frac{G_{\max}}{24 N^2}. \tag{33}$$

Finally, we use our stronger assumption to determine the bound for $G_{\max} = \max |g''(s)|$. We have:

$$g''(s) = (\Delta E)^T \left( \nabla^3 \mathcal{L}(X(s))[\Delta E] \right) \Delta E. \tag{34}$$

Using the bound on the norm of the third-order derivative tensor, $\left\|\nabla^3 \mathcal{L}(X)\right\| \leq K$:

$$|g''(s)| \leq \left\|\nabla^3 \mathcal{L}(X(s))\right\| \cdot \|\Delta E\|^3 \leq K \|\Delta E\|^3. \tag{35}$$

Thus, $G_{\max} \leq K \|\Delta E\|^3$.

Substituting the upper bound for $G_{\max}$ into the total error formula, we could get:

$$\text{Total Error} \leq \frac{K \|\Delta E\|^3}{24 N^2}. \tag{36}$$

It shows that the error bound for the composite midpoint rule is: $\mathcal{O}\left(\frac{1}{N^2}\right)$. In practice, even using the right-endpoint method often yields satisfactory results.

## C    The Maximum Coverage for Training Data Batches

### C.1    The Monotone Submodular Function

**Proposition 3.** *The feature coverage function $F(\mathcal{S}) = |\mathcal{F}(\mathcal{S})|$ is a monotone submodular function.*

*Proof.* Let $\mathcal{F}(\mathcal{S})$ denote the set of unique feature values covered by batch set $\mathcal{S}$. For any $\mathcal{A} \subseteq \mathcal{B} \subseteq \mathcal{S}$ and a new batch $b \in \mathcal{S} \setminus \mathcal{B}$, we have:

$$F(\mathcal{A} \cup \{b\}) - F(\mathcal{A}) = |\mathcal{F}(\mathcal{A} \cup \{b\}) \setminus \mathcal{F}(\mathcal{A})|$$
$$F(\mathcal{B} \cup \{b\}) - F(\mathcal{B}) = |\mathcal{F}(\mathcal{B} \cup \{b\}) \setminus \mathcal{F}(\mathcal{B})|$$

Since $\mathcal{A} \subseteq \mathcal{B}$, we have $\mathcal{F}(\mathcal{A}) \subseteq \mathcal{F}(\mathcal{B})$. The marginal gain from adding $b$ decreases as:

$$|\mathcal{F}(\{b\}) \setminus \mathcal{F}(\mathcal{A})| \geq |\mathcal{F}(\{b\}) \setminus \mathcal{F}(\mathcal{B})|, \tag{37}$$

proving submodularity. Monotonicity follows from $\mathcal{F}(\mathcal{S}) \subseteq \mathcal{F}(\mathcal{S} \cup \{b\})$. $\qquad\square$

Therefore, from (Nemhauser et al., 1978), the greedy selection strategy in Equation 4 achieves a **(1 - 1/e)**-approximation guarantee for maximizing the feature coverage function. Our batch selection process with termination condition $|\mathcal{S}'|/|\mathcal{S}| > \rho$ corresponds to a cardinality constraint $k = \lceil \rho|\mathcal{S}|/B \rceil$. The iterative selection of batches with maximum marginal gain exactly implements the classical greedy algorithm. This theoretical guarantee ensures that MFCS provides near-optimal feature coverage while maintaining the efficiency of greedy selection. The bitmap optimization further makes it practical for large-scale datasets.

### C.2    The Choice of $\rho$

In the main text, $\rho$ represents the ratio of selected batches to all batches, which essentially corresponds to the proportion of selected samples. Since recommender system datasets are typically large, it is feasible to approximate within an acceptable error range using a small subset of samples. The subsequent idea of maximizing feature value coverage is also based on these selected samples.

We aim to select an appropriate sampling ratio $\rho$ to achieve maximum coverage while keeping the distribution bias within an acceptable range. The *Dvoretzky-Kiefer-Wolfowitz* (DKW) inequality establishes a direct relationship between the maximum deviation between the empirical and true distributions and both the sample size and sampling method of the selected dataset $\mathcal{S}'$. Specifically, it states:

$$P\left(\sup_x |\Phi_n(x) - \Phi(x)| > \epsilon\right) \leq 2e^{-2n\epsilon^2}, \tag{38}$$

where $\Phi$ is the cumulative distribution function (CDF), and $\Phi_n$ is the empirical CDF based on $n$ samples. This result implies that as the sample size $n$ increases, the probability bound decreases, and the empirical distribution better approximates the true distribution. A smaller allowed deviation $\epsilon$

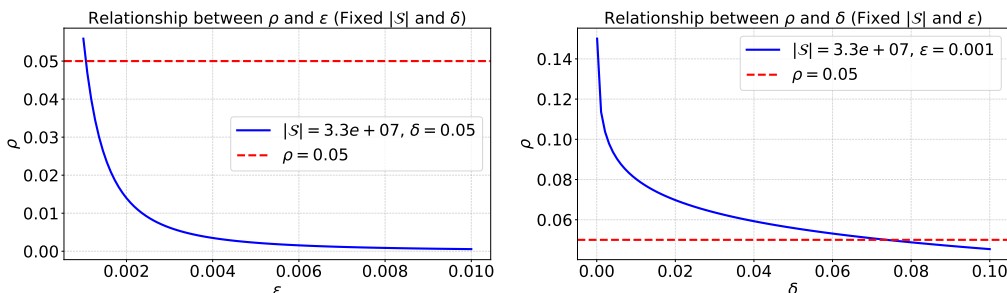

Figure 9: Analysis of the relationship between $\rho$ and $\epsilon$.

Figure 10: Analysis of the relationship between $\rho$ and $\delta$.

requires a larger $n$ to maintain the same confidence level. The core conclusion of the DKW inequality states that the probability upper bound for the maximum deviation exceeding a threshold $\epsilon$ decreases exponentially as the sample size $n = |\mathcal{S}'| = \rho|\mathcal{S}|$ increases.

Based on this, if we set $\delta$ as the confidence level, and constrain $2e^{-2\rho|\mathcal{S}|\epsilon^2} \leq \delta$, we can derive an analytical solution for the required sampling ratio $\rho$ that satisfies the given confidence level while minimizing the threshold, as:

$$\rho \geq \frac{1}{2\epsilon^2|\mathcal{S}|} \ln \frac{2}{\delta}. \tag{39}$$

Generally, we set $\delta \leq 0.05$, which ensures a confidence level of at least 95%, and aim to keep the distribution deviation $\epsilon$ as small as possible (*e.g.*, $1.0 \times 10^{-3}$). For the Avazu dataset, substituting $|\mathcal{S}| \approx 3.3 \times 10^7$ gives $\rho_{\min} \approx 0.05$. For other datasets, the same method can be applied to determine the corresponding value. To maintain consistency, we set $\rho = 0.05$ for all datasets, which yields satisfactory results.

More intuitively, we fix the relevant parameters and show the relationships between the sampling ratio $\rho$ and the bias threshold $\epsilon$ (in Figure 9), as well as between the sampling ratio $\rho$ and the confidence level $\delta$ (in Figure 10).

## D  EXPERIMENT DETAILS

### D.1  DATASET

We use four benchmark datasets in our experiment.

- **iPinYou**[6]: This dataset originates from the iPinYou Global RTB Bidding Algorithm Competition, held in 2013 across three distinct seasons. It includes comprehensive training datasets and leaderboard testing datasets for each season, covering DSP bidding, impression, click, and conversion logs. The dataset for final evaluation is withheld by iPinYou and is reserved for testing purposes.

- **Ali-CCP**[7]: Collected from the recommendation system logs of the mobile Taobao app, this dataset ensures that the training data precedes the test data. It is divided into three parts: Sample ID Section, Labels Section, and Features Section, with a total of 23 feature fields. The dataset comprises over 80 million records for both training and testing.

- **Avazu**[8]: This dataset is used in the Kaggle CTR prediction competition and contains nearly 40 million interaction records across 22 fields. Based on prior preprocessing steps, the *hour* field has been subdivided into three separate fields: *weekday*, *weekend*, and a newly defined *hour* field, resulting in a total of 24 fields.

---

[6]https://contest.ipinyou.com/
[7]https://tianchi.aliyun.com/dataset/408
[8]https://www.kaggle.com/c/avazu-ctr-prediction/

- **Criteo[9]:** Serving as a benchmark dataset in the real world, Criteo contains approximately 45 million records of user clicks, with all fields anonymized. The dataset includes 39 fields, comprising 26 categorical features and 13 numerical features. Following the methods of earlier studies (Wang et al., 2022), we convert the numerical features into categorical ones[10].

For the Ali-CCP dataset, the source website provides both training and test sets. Since the test set occurs after the training set, directly splitting the validation set from the training data could lead to data leakage. Therefore, we split half of the test set to form the validation set, ensuring that both the validation and test sets are disjoint and occur after the training data.

## D.2 BASELINES

For the baselines, we select two types of feature selection methods. For field selection, we choose six representative methods: Random Forest (Breiman, 2001), XGBoost (Chen & Guestrin, 2016), Recursive Feature Elimination (RFE) (Chen & Jeong, 2007), Permutation Feature Importance (PFI) (Fisher et al., 2019), AdaFS (Lin et al., 2022), and MvFS (Lee et al., 2023). Except for the last two, which use neural networks and learn sample-wise field weights through a controller, all other methods are traditional machine learning techniques. For feature value selection, we primarily compare OptFS (Lyu et al., 2023) and two variants of the EffSelect method as our approaches.

- **Random Forest (RF) (Breiman, 2001):** Random Forest is an ensemble learning method that constructs a multitude of decision trees during training and outputs the class that is the mode of the classes (classification) or mean prediction (regression) of the individual trees. It works by creating multiple decision trees from bootstrapped samples of the training data, reducing overfitting and improving generalization. Each tree is trained on a random subset of features, helping to minimize bias and variance. Random Forest is well-known for its robustness and ability to handle large datasets with higher accuracy.

- **XGBoost (Chen & Guestrin, 2016):** XGBoost (Extreme Gradient Boosting) is a highly efficient and scalable implementation of gradient boosting. It builds an ensemble of decision trees sequentially, where each new tree corrects the errors made by the previous ones. The algorithm uses a regularization term to control model complexity, helping to prevent overfitting. Feature importance in XGBoost is computed based on how frequently a feature is used to split a node and the gain it contributes to improving the model's predictive performance. XGBoost has become popular due to its speed, accuracy, and ability to handle various data types and missing values.

- **Recursive Feature Elimination (RFE) (Guyon et al., 2002):** Recursive Feature Elimination is a wrapper-based feature selection technique that recursively removes the least important features from the model. The process involves:

  1. Training a model and evaluating the importance of each feature, typically using feature weights (for example, the coefficients in linear models or feature importance scores in tree-based models).
  2. Iteratively eliminating the features with the lowest importance scores.
  3. Repeating the process until the desired number of features is selected, ensuring that only the most significant features are retained.

  In our experiment, we use RFE based on Linear Regression.

- **Permutation Feature Importance (PFI) (Fisher et al., 2019):** Partial Feature Importance (PFI) is a technique used to evaluate the importance of each feature by assessing how much its value affects the predictive performance of the model. The method works by shuffling the values of a specific feature across the dataset and observing the resulting change in the model's performance. If the predictive performance degrades significantly after shuffling a feature, it indicates that the feature plays an important role in the model's decision-making. PFI is particularly useful for identifying key features when dealing with complex, high-dimensional datasets and is widely applied in both supervised learning and model explainability.

---

[9]https://www.kaggle.com/c/criteo-display-ad-challenge/
[10]https://www.csie.ntu.edu.tw/r01922136/kaggle-2014-criteo.pdf

- **AdaFS (Lin et al., 2022):** AdaFS (Adaptive Feature Selection) is a feature selection method that employs a controller network to dynamically select the most informative features during the training process. This approach uses sample-wise learning to adaptively choose the subset of features that maximizes model performance.

- **MvFS (Lee et al., 2023):** MvFS (Multi-view Feature Selection) is an enhancement of AdaFS that introduces the concept of multi-view controllers for feature selection. In this method, multiple controllers operate in parallel, each focusing on a different "view" of the data. This approach helps capture diverse feature interactions across different data modalities or subsets, making the feature selection process more robust. MvFS improves upon AdaFS by allowing the model to adaptively select features across multiple perspectives, leading to better performance in tasks where feature dependencies vary across different views or contexts.

In addition, we also included several Autoencoder-based feature selection methods as baselines, namely Concrete Autoencoder (CAE) (Balın et al., 2019), IP-CAE (Indirectly Parameterized CAE) (Nilsson et al., 2024), and L2X (Chen et al., 2018).

- **Concrete Autoencoder (CAE) (Balın et al., 2019):** Concrete Autoencoder is a **feature field selection** method, where the core idea is to construct an "encoder-decoder" structure. The encoder serves as a differentiable, learnable "soft" selection gate, while the decoder uses the selected feature fields to perform a task. By optimizing the final loss of the task, the "selection gate" is backpropagated to learn how to pick the most informative feature fields. Specifically, for each batch of data: $E \in \mathbb{R}^{B \times M \times D}$, where $B$ is the batch size, $M$ is the total number of feature fields, and $D$ is the embedding size of each feature field, we define the learnable importance weights as

$$\boldsymbol{\alpha} \in \mathbb{R}^{k \times M},$$

representing which $k$ feature fields are selected (with each row approximating a one-hot encoding). Applying Gumbel-Softmax to this weight matrix yields $W \in \mathbb{R}^{k \times M}$.

- **Indirectly Parameterized CAE (IP-CAE) (Nilsson et al., 2024):** This is an improved version of the original CAE method. Specifically, IP-CAE modifies the logic of the weights $\boldsymbol{\alpha}$ by changing it to an indirect parameterization. While the traditional CAE requires learning $k \times M$ parameters to determine feature selection, IP-CAE only needs to learn $k \times dim_{ip}$ intrinsic parameters, which are then mapped to importance values through a mapping network. There are four different design patterns for this mapping network:

  - **Shared mode (IP-Share):** All $k$ latent vectors share the same mapping network, which reduces the complexity of the network. The code implementation is as follows:

    ```
    self.mapping_network = self._create_mlp(net_dims)
    ```

  - **Separate mode (IP-Sep):** Each latent vector is assigned a separate mapping network, meaning each selection vector has its own mapping function. The implementation is as follows:

    ```
    self.mapping_networks = nn.ModuleList([self.
        _create_mlp(net_dims) for _ in range(self.
        num_select)])
    ```

  - **FC mode (IP-FC):** All latent vectors are flattened and passed through a larger fully connected layer, which can capture more complex nonlinear features. The corresponding code implementation is:

    ```
    self.mapping_network = self._create_mlp(net_dims)
    ```

  - **Diag mode (IP-Diag):** A special diagonal or scalar network replaces the standard linear layer, and the learnable parameters are constrained to the elements on the diagonal or a single scalar value. The implementation is as follows:

    ```
    self.mapping_network = DiagNet(self.ip_dim)
    ```

---

**Algorithm 1:** Maximizing Feature Coverage Sampling (MFCS)

---

**Input:** Set of all disjoint data batches $\mathcal{B} = \{\mathcal{B}_1, \ldots, \mathcal{B}_m\}$, Number of batches ratio $\rho$
**Output:** The set of selected batches $\mathcal{S}'$
```
// Initialization
```
$\mathcal{S}' \leftarrow \emptyset, \mathcal{V}^* \leftarrow \emptyset, \mathcal{B}_{rem} \leftarrow \mathcal{B}.$
```
// Iteratively select k batches in a greedy manner
```
**while** *true* **do**
    ```// Terminate if no more batches are available```
    **if** $\mathcal{B}_{rem} = \emptyset$ **then**
    **end**
    ```// Find the best candidate batch to add next```
    $\mathcal{B}_{\text{best}} \leftarrow \arg\max_{\mathcal{B}_b \in \mathcal{B}_{rem}} |\mathcal{F}(\mathcal{B}_b) \setminus \mathcal{V}^*|$ ;
    ```// Pre-check: Break if adding this batch would exceed the```
    ```   ratio```
    **if** $(|\mathcal{S}' \cup \{\mathcal{B}_{best}\}|) > \rho \times |\mathcal{S}|$ **then**
    **end**
    ```// If the check passes, commit to adding the batch```
    $\mathcal{S}' \leftarrow \mathcal{S}' \cup \{\mathcal{B}_{\text{best}}\}$ ;
    $\mathcal{V}^* \leftarrow \mathcal{V}^* \cup \mathcal{F}(\mathcal{B}_{\text{best}})$ ;
    $\mathcal{B}_{\text{rem}} \leftarrow \mathcal{B}_{\text{rem}} \setminus \{\mathcal{B}_{\text{best}}\}$ ;
**end**
**return** $\mathcal{S}'$

---

- **L2X (Chen et al., 2018):** L2X is a finer-grained feature selection method. Unlike Balın et al. (2019) and Nilsson et al. (2024), which assign the same feature importance across all samples, L2X uses an explainer network to compute feature importance scores for each individual sample.

### D.3 GENERAL HYPERPARAMETERS

The MaskNet model is configured for binary classification with a binary cross-entropy loss function, using the Adam optimizer with a learning rate of 1e-3. It employs a batch size of 10,000, an embedding dimension of 8, and a DNN architecture with three hidden layers of 400 units each, activated by ReLU. Regularization is disabled, and the model includes layer normalization for both the embedding and DNN layers. The model is trained for 100 epochs with AUC as the monitoring metric and a seed value of 20242025, using a shuffle strategy and a 50% feature retention ratio. Pretraining is required, and the model is set for retraining in autofeat mode, with parallel block processing using 1 block of 64 dimensions.

Following previous research (Jia et al., 2024), we set the embedding size to 8 for all models. The batch size is set to 10,000 for faster training. The learning rate is set globally to 1.0e-3. In this study, an early stopping strategy was employed, where the model would terminate training prematurely if the AUC-Logloss on the validation set did not improve over two consecutive training epochs. Additionally, common optimization techniques such as the Adam optimizer (Kingma & Ba, 2015) and the Xavier initialization method (Glorot & Bengio, 2010) were utilized.

### D.4 FEATURE VALUE IMPORTANCE CALCULATION

See in Algorithm 2.

## E ADDITIONAL EXPERIMENTS

### E.1 COMPARE WITH AUTOENCODER-BASED SELECTION METHODS

As shown in Table 4, for the autoencoder-based feature selection method, the latent **selection embedding** is set to 32, and additional experiments are conducted using MaskNet. It can be observed

---

**Algorithm 2:** Feature Value Importance Calculation Algorithm

---

**Input:** Pre-trained embedding table and model $\Theta_{\mathcal{S}'}$, validation dataset $\mathcal{S}_{\text{val}}$, point number $N$, kept feature values ratio $\alpha$.

**Output:** Final selected feature values.

Initialize starting point embeddings $E^*$ using field-wise means:

**for** *each feature field $f$* **do**

    $F_f \leftarrow \{v \text{ belongs to field } f\}$.

    $E_v^* \leftarrow \frac{1}{|F_f|} \sum_{v_i \in F_f} E_{v_i}, \ \forall v \in F_f$ ;          // Eq.14

    $E^* = \text{concat}(E_v^*)$.

**end**

Initialize importance scores $I_v$.

**for** *each batch $\mathcal{S}'_{val} \in \mathcal{S}_{val}$* **do**

    **for** $k \leftarrow 1$ *to* $N$ **do**

        Get current embeddings $E$ from trained parameters $\Theta_{\mathcal{S}'}$.

        Compute delta embeddings $\Delta E \leftarrow \frac{E - E^*}{N}$.

        $E^{(k)} \leftarrow E^* + k\Delta E$.

        **Forward pass**: $\mathcal{L} \leftarrow \mathcal{L}_{\mathcal{S}'_{\text{val}}}(E^{(k)})$.

        **Backward pass**: Compute $\nabla_{E^{(k)}} \mathcal{L}$.

        **for** *each feature value $v$ in batch* **do**

            $I_v \leftarrow I_v + \left| \nabla_{E_v^{(k)}} \mathcal{L} \cdot \Delta E_v \right|$.

        **end**

    **end**

**end**

Sort all $v$ by $I_v$ in descending order.

Select the $v$ in top-$\alpha$ importance.

**return** *Selected feature values*

---

Table 4: Comparison of datasets with different feature selection methods on MaskNet.

| Model | Dataset | Metric | Base | CAE | IP-Share | IP-Sep | IP-FC | IP-Diag | L2X | EffSelect$_{\mathcal{Z}}$ | EffSelect$_{\mathcal{M}}$ |
|-------|---------|--------|------|-----|----------|--------|-------|---------|-----|-----------|-----------|
| MaskNet | Criteo | AUC | 0.8098 | 0.8001 | 0.8014 | 0.8023 | 0.8048 | 0.8011 | 0.8041 | 0.8110 | **0.8111** |
| | | Logloss | 0.4420 | 0.4509 | 0.4498 | 0.4489 | 0.4467 | 0.4498 | 0.4471 | 0.4408 | **0.4407** |
| | Avazu | AUC | **0.7914** | 0.7865 | 0.7882 | 0.7881 | 0.7896 | 0.7864 | 0.7892 | 0.7757 | 0.7766 |
| | | Logloss | **0.3731** | 0.3764 | 0.3749 | 0.3752 | 0.3742 | 0.3763 | 0.3744 | 0.3824 | 0.3816 |
| | iPinYou | AUC | 0.7674 | 0.7496 | 0.7505 | 0.7532 | 0.7629 | 0.7477 | 0.7541 | 0.7683 | **0.7699** |
| | | Logloss | 0.5608 | 0.5671 | 0.5664 | 0.5760 | 0.5623 | 0.5681 | 0.5631 | 0.5598 | **0.5581** |
| | Ali-CCP | AUC | 0.6056 | 0.5978 | 0.6013 | 0.5960 | 0.6029 | 0.6055 | 0.6072 | 0.6010 | **0.6109** |
| | | Logloss | 0.1637 | 0.1631 | 0.1633 | 0.1637 | 0.1635 | 0.1635 | 0.1638 | 0.1624 | **0.1619** |

that EffSelect achieves the best prediction results on three out of four datasets, with only 10% of the feature values. Among the baselines, the L2X method performs the best, as it applies feature weighting on a sample-wise basis. On the other hand, IP-CAE uses an indirect network to model feature importance, and the increased number of parameters intuitively allows better modeling of the importance between different feature fields, resulting in better performance compared to CAE.

E.2   THE INFLUENCE OF HYPERPARAMETER $\rho$.

We vary $\rho$ for pre-training within $\{0.05, 0.1, 0.5, 1\}$, where $\rho = 1$ corresponds to using all the training data. As shown in Figure 11, using more training data does not necessarily improve feature value selection results.

For example, on Criteo, the final selection results with 5% and 100% of the training data are almost the same. When re-training with 5% of the data at $\alpha = 0.1$, the performance is even better than

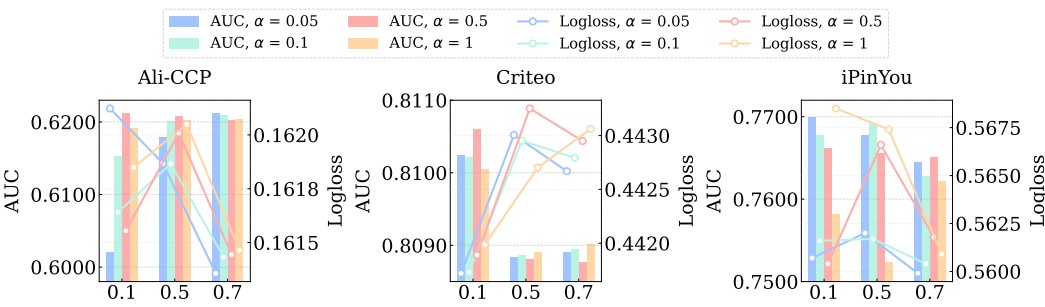

Figure 11: Bar charts and line charts showing the variation of $\alpha$ across three datasets as $\rho$ changes. The bar charts represent AUC, while the line charts depict Logloss.

with full training. This also occurs on iPinYou, possibly because more training data introduces more noise, leading to worse performance. On Ali-CCP, as $\rho$ increases, smaller values of $\alpha$ yield more significant prediction results. This could be because with more data, top feature values are better able to distinguish from noisy features. However, when $\alpha = 0.7$, the result from training with full data is worse than that from training with batches obtained through MFCS.

### E.3 DATASET COVERAGE RATIO $\rho$ AND PRETRAINING LOSS

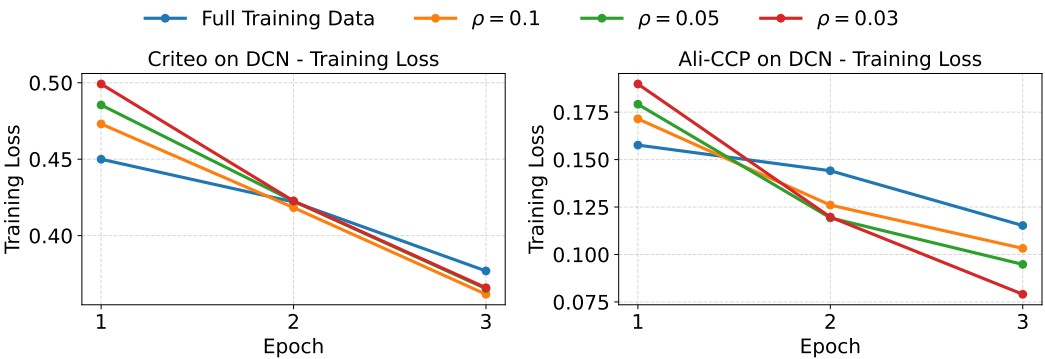

Figure 12: Training loss for Criteo and Ali-CCP datasets on DCN under different dataset coverage ratios $\rho$.

In this subsection, we analyze how the Logloss on the training set changes over training epochs. We use DCN as the backbone model and conduct experiments on the Criteo and Ali-CCP datasets. Since we apply an early stopping strategy—training stops when the AUC-Logloss on the validation set does not decrease for two consecutive epochs—the total number of training steps is relatively small.

As shown in Figure 12, the overall decreasing trend of Logloss during pretraining using different values of $\rho$ selected by the MFCS method is similar to that of training on the full dataset. This indicates that the selected data batches effectively guide gradient descent. However, their specific effectiveness differs slightly. In particular, after the first epoch, the Logloss from MFCS-selected batches is slightly higher than that of using the full training data. But as training continues, models trained on MFCS batches fit the data better than those trained on the full dataset.

Moreover, larger $\rho$ values tend to result in lower Logloss in the first epoch, reflecting that data selected by MFCS can lead to more stable and robust training and convergence results.

### E.4 CASE STUDY

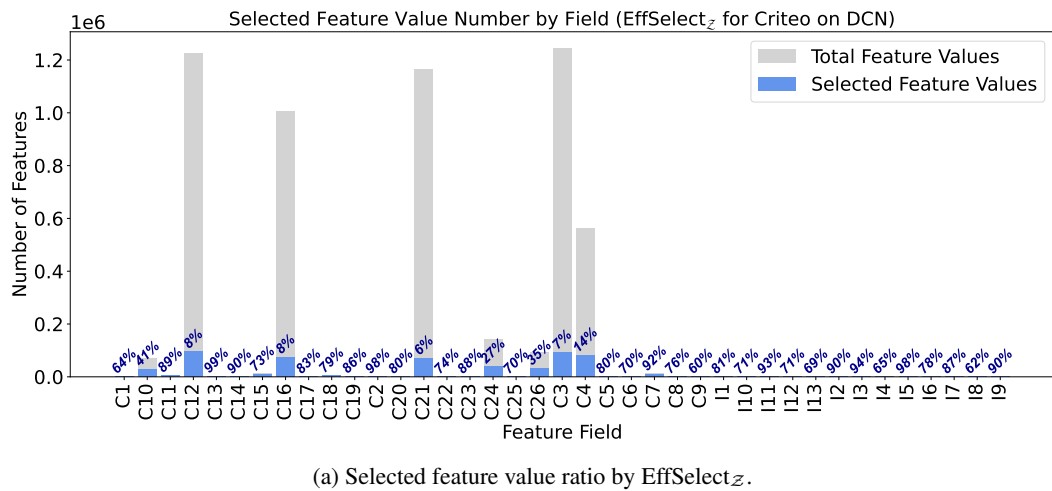

(a) Selected feature value ratio by EffSelect$_{\mathcal{Z}}$.

(b) Selected feature value ratio by EffSelect$_{\mathcal{M}}$.

Figure 13: Case study for selected feature values with Criteo on DCN.

We conduct a case study on the selected feature values. Specifically, we focus on the Criteo dataset with DCN and compare the feature values selected by the two variants, EffSelect$_{\mathcal{Z}}$ and EffSelect$_{\mathcal{M}}$. Figure 13a shows the results for EffSelect$_{\mathcal{Z}}$, while Figure 13b shows those for EffSelect$_{\mathcal{M}}$.

From the figures, we could observe that for the Criteo dataset, most of the filtered feature values come from fields with a large number of unique values. This is intuitive, as such fields often contain more noise. Without strong support from golden samples, these values are less likely to contribute positively during training. In contrast, fields with fewer unique values also have some values filtered out, but a higher proportion of values are retained. This may be because these values have more reliable samples, which help improve generalization.

When comparing the selected feature values between EffSelect$_{\mathcal{Z}}$ and EffSelect$_{\mathcal{M}}$, we find that for fields with many values, both methods yield similar selection ratios. However, for fields with fewer values, such as C15 and C26, the selection ratios differ significantly. This indicates that using mean embedding (for values within the same field) as the starting point for Taylor expansion affects feature selection more in low-cardinality fields, compared to using zero embedding.

### E.5    BITMAP IMPLEMENTATION

As an efficient data structure, a bitmap is essentially an array where each element is a binary state (0 or 1), using only one bit of memory. This design is ideal for compactly storing variables with a small number of possible states (e.g., two states) in a sequential array of bits—precisely the structure of

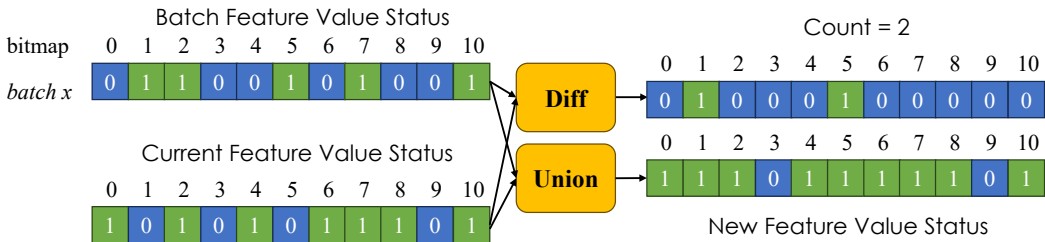

Figure 14: Diagram of `bitmap` with 0-10 feature values.

feature value indices. Such bitmaps allow for fast operations like lookup, deduplication, and counting across large sets of values. Moreover, bitmaps support quick set operations such as intersection, union, and difference, which are useful for relational queries between datasets.

Figure 14 shows an example of using a bitmap to count and update feature values. Here, 0 indicates the presence of a feature value at the given index, while 1 means it is absent. The top-left `bitmap` is `01100101001`, representing that feature values at indices 1, 2, 5, 7, and 10 are present in the current batch. The bottom-left `bitmap` tracks feature values already seen across previous batches. For each new batch, we compute the number of new feature values using a `Diff` operation—counting the number of 1s in the top-right `bitmap`. We then use a `Union` operation to update the set of known feature values before moving to the next batch. This process continues until the selected batch ratio reaches $\rho$. Since bitwise operations are highly efficient on modern hardware, this approach is much faster than using normal arrays or maps to track feature values.

### E.6 DETAILED EFFICIENCY ANALYSIS

In this section, we provide the parameter counts during the retraining stage for all methods and datasets used in the main experiments. As shown in Figure 15, EffSelect achieves the lowest parameter count during retraining. On average, the parameter count is only 15% of the original model. Compared to the 10% of features selected, the additional 5% comes from the model's architecture. This analysis further highlights the practical value of EffSelect.

## F LLM USAGE STATEMENT

The LLM was used as a tool to assist with polishing the writing and did not directly contribute to the research findings or results. All content generated by the LLM was thoroughly reviewed and edited by the authors to ensure its relevance, accuracy, and scientific integrity.

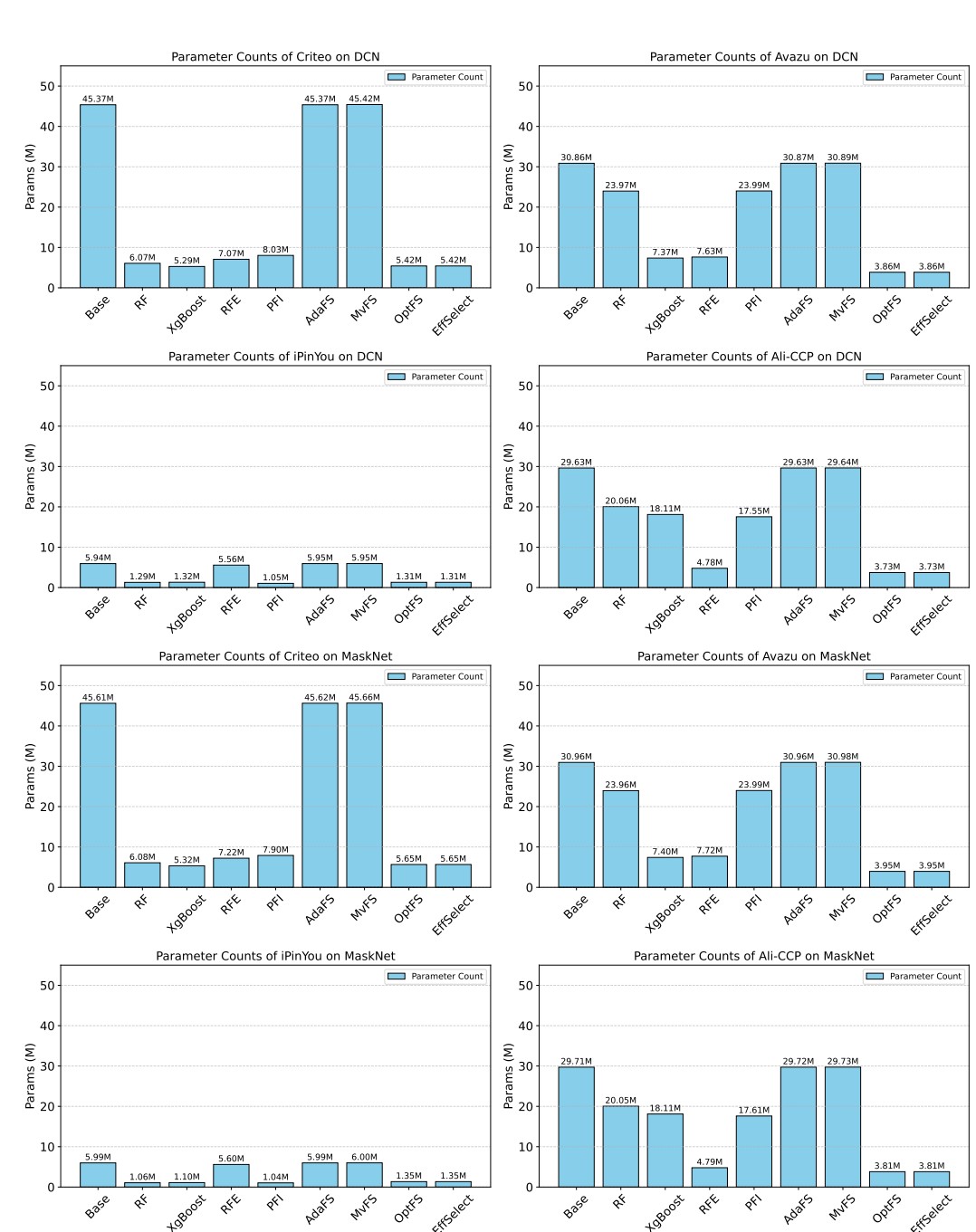

Figure 15: Parameter counts for different methods when retraining.

