# OpenReview forum: "EffSelect: Efficient Feature Value Selection for Deep Recommender Systems with Mini-Batch Training"
_ICLR.cc/2026/Conference — Submitted to ICLR 2026_

### Official Review · Reviewer_sTD5 · 2025-10-30

**Soundness:** 3
**Presentation:** 3
**Contribution:** 3
**Rating:** 6
**Confidence:** 3

**Summary:**

This paper introduces EffSelect, a novel framework for feature value selection in deep recommender systems. The method operates at the feature value level rather than the feature field level, aiming to reduce redundancy and noise in embedding tables without introducing additional learnable parameters. The approach combines a mini-batch pre-training strategy (MFCS) with a gradient-based importance scoring mechanism (FeatIS) to efficiently identify and retain only the most informative feature values. Extensive experiments on four public datasets demonstrate that EffSelect outperforms existing feature selection methods in both performance and efficiency.

**Strengths:**

1.EffSelect addresses a critical gap by performing feature value-level selection, which is finer-grained than field-level selection and more practical than gate-based value selection methods.
2.The method does not introduce additional learnable parameters, making it lightweight and suitable for dynamic recommender systems.
3.The use of numerical integration and gradient-based importance estimation (FeatIS) is well-motivated and theoretically justified, with error bounds provided in the appendix.
4.Requires only 5% of training data for warm-up, making it highly efficient and suitable for large-scale systems.
5.Experiments are conducted on four diverse datasets and two base models (DCN and MaskNet), with comparisons against a wide range of field-level and value-level baselines.

**Weaknesses:**

1.The method assumes a static dataset for pre-training and selection. In real-world systems, new feature values may emerge over time. A discussion or experiment on incremental or online adaptation of the feature set would strengthen the practicality claim.
2.The performance is sensitive to α (selection ratio), and the optimal α varies across datasets. This may require tuning in practice. The authors could propose a heuristic or adaptive method for setting α based on dataset characteristics.
3. While the paper compares with many strong baselines, some recent feature selection or embedding pruning methods are not included.
4. The description of the bitmap implementation in MFCS is somewhat brief. A more detailed pseudocode or example would help reproducibility.
5. The authors could explore adaptive N or early stopping strategies to reduce the number of gradient evaluations.

**Questions:**

1.The method assumes a static dataset for pre-training and selection. In real-world systems, new feature values may emerge over time. A discussion or experiment on incremental or online adaptation of the feature set would strengthen the practicality claim.
2.The performance is sensitive to α (selection ratio), and the optimal α varies across datasets. This may require tuning in practice. The authors could propose a heuristic or adaptive method for setting α based on dataset characteristics.
3. While the paper compares with many strong baselines, some recent feature selection or embedding pruning methods are not included.
4. The description of the bitmap implementation in MFCS is somewhat brief. A more detailed pseudocode or example would help reproducibility.
5. The authors could explore adaptive N or early stopping strategies to reduce the number of gradient evaluations.

---

> ### Author Response · Authors · 2025-11-25
> **Response for Reviewer sTD5 (1/3)**
>
> We sincerely appreciate your valuable comments and suggestions. They are critical to improving the overall quality of our paper.
>
> ---
>
> ### W1 & Q1: On Dynamic Datasets
>
> You raised an important point, and in fact, this is one of the key advantages of **EffSelect**—its ability to rapidly adapt to temporal distribution shifts.
>
> In real-time recommender systems, the total number of feature values is usually very large. Even as the data distribution evolves over time, the majority of core feature values often remain unchanged. Therefore, by leveraging our **MFCS**, we can efficiently perform mini-batch training and quickly re-evaluate the importance of feature values to adapt to the new data scenario.
>
> ---
>
> ### W2 & Q2: On the Selection of $\alpha$
>
> In **EffSelect**, the hyperparameter $\alpha$ controls the proportion of selected feature values, and can be flexibly adjusted to suit different data scenarios and value importance levels.
>
> As stated in Lines 373–374 of the main paper, we have **benchmarked our method and baselines across two backbone models and four datasets**. In all main experiments, we **fixed $\alpha$ at a 10% ratio**, meaning we used only 10% of feature values. Even under this constraint, our method **outperforms or matches full-feature training in most cases**. Therefore, it is **not necessary to carefully select $\alpha$ per dataset**, nor did we tune $\alpha$ across datasets in our main results.
>
> Regarding the **optimal choice of $\alpha$**, as demonstrated in **Fig. 11 (Page 22)**, some datasets can further benefit from more precise tuning based on validation performance. Fortunately, such tuning is **efficient under EffSelect**.
>
> Moreover, we provide two practical heuristics to guide the choice of $\alpha$:
> 1. Datasets with more feature values tend to contain more noise and redundancy, and thus may benefit from lower $\alpha$.
> 2. The relationship between performance and $\alpha$ often forms a **quasi-convex** shape—initially increasing (as feature values are better trained) and then decreasing (due to noise being introduced). If this assumption holds, one can employ a **ternary search** over the continuous domain to optimize $\alpha$. Alternatively, **heuristic methods such as simulated annealing** may also help guide the search.

---

> ### Author Response · Authors · 2025-11-25
> **Response for Reviewer sTD5 (2/3)**
>
> ### W3 & Q3: Additional Baselines
>
> Thank you for your insightful comment. In our **initial submission**, we have already included a variety of recent or popular feature selection methods in **Appendix E.1**, including Autoencoder-based approaches. We invite you to refer to that section directly.
>
> As for **embedding pruning** methods, our focus is on optimizing at the **feature (value) level** rather than on compressing model parameters. Therefore, the baselines we selected align with those suggested by Reviewer Wiy9, and target feature field/value selection more directly.
>
> Nevertheless, we have now included **LPFS** [1] as an additional baseline for comparison. Below are the experimental results:
>
> #### Add LPFS on DCN:
>
> | Dataset    | Metric  | DCN-Base     | RF     | Xgboost | RFE    | PFI    | AdaFS  | MvFS   | OptFS  | **LPFS** | EffSelect-zero | EffSelect-mean |
> |------------|---------|--------------|--------|---------|--------|--------|--------|--------|--------|----------|----------------|----------------|
> | **Criteo** | AUC     | 0.8090       | 0.7879 | 0.7793  | 0.8058 | 0.8034 | 0.7998 | 0.7996 | 0.8077 | 0.7979   | **0.8102**     | **0.8102**     |
> |            | Logloss | 0.4427       | 0.4610 | 0.4674  | 0.4457 | 0.4478 | 0.4514 | 0.4525 | 0.4439 | 0.4525   | 0.4418         | **0.4417**     |
> | **Avazu**  | AUC     | **0.7908**   | 0.7076 | 0.7500  | 0.7717 | 0.7634 | 0.7823 | 0.7836 | 0.7877 | 0.7633   | 0.7744         | 0.7737         |
> |            | Logloss | **0.3735**   | 0.4134 | 0.3953  | 0.3843 | 0.3880 | 0.3829 | 0.3830 | 0.3760 | 0.3881   | 0.3829         | 0.3829         |
> | **iPinYou**| AUC     | 0.7642       | 0.7383 | 0.7635  | 0.7319 | 0.7572 | 0.7391 | 0.7270 | 0.7624 | 0.7621   | 0.7683         | **0.7699**     |
> |            | Logloss | 0.5630       | 0.5766 | 0.5656  | 0.5894 | 0.5662 | 0.5821 | 0.5988 | 0.5623 | 0.5618   | 0.5620         | **0.5607**     |
> | **Ali-CCP**| AUC     | 0.5956       | 0.5762 | 0.5834  | 0.5743 | 0.5939 | 0.6004 | 0.6009 | 0.5979 | 0.5949   | 0.6000         | **0.6021**     |
> |            | Logloss | 0.1639       | 0.1631 | 0.1630  | 0.1640 | 0.1631 | 0.1656 | 0.1656 | 0.1644 | 0.1630   | 0.1622         | **0.1621**     |
>
> #### Add LPFS on MaskNet:
>
> | Dataset    | Metric  | MaskNet-Base | RF     | Xgboost | RFE    | PFI    | AdaFS  | MvFS   | OptFS  | **LPFS**  | EffSelect-zero | EffSelect-mean |
> |------------|---------|--------------|--------|---------|--------|--------|--------|--------|--------|-----------|----------------|----------------|
> | **Criteo** | AUC     | 0.8098       | 0.7880 | 0.7722  | 0.8062 | 0.8009 | 0.7999 | 0.7999 | 0.8086 | 0.7805    | 0.8110         | **0.8111**     |
> |            | Logloss | 0.4420       | 0.4609 | 0.4728  | 0.4453 | 0.4501 | 0.4509 | 0.4511 | 0.4431 | 0.4667    | 0.4408         | **0.4407**     |
> | **Avazu**  | AUC     | **0.7914**   | 0.7129 | 0.7506  | 0.7724 | 0.7646 | 0.7834 | 0.7849 | 0.7900 | 0.7641    | 0.7757         | 0.7766         |
> |            | Logloss | **0.3731**   | 0.4390 | 0.3950  | 0.3848 | 0.3876 | 0.3816 | 0.3808 | 0.3741 | 0.3880    | 0.3824         | 0.3816         |
> | **iPinYou**| AUC     | 0.7674       | 0.7242 | 0.7666  | 0.7534 | 0.7563 | 0.7580 | 0.7653 | 0.7570 | 0.7618    | 0.7683         | **0.7699**     |
> |            | Logloss | 0.5608       | 0.5726 | 0.5628  | 0.5624 | 0.5646 | 0.5684 | 0.5629 | 0.5622 | 0.5698    | 0.5598         | **0.5581**     |
> | **Ali-CCP**| AUC     | 0.6056       | 0.5739 | 0.5815  | 0.5733 | 0.5986 | 0.6020 | 0.5992 | 0.6005 | 0.5940    | 0.6010         | **0.6109**     |
> |            | Logloss | 0.1637       | 0.1636 | 0.1630  | 0.1641 | 0.1660 | 0.1651 | 0.1661 | 0.1650 | 0.1640    | 0.1624         | **0.1619**     |
>
> Again, we observe that **LPFS underperforms relative to EffSelect**, further reinforcing the strength of our proposed method.
>
> [1] Guo, Yi, et al. "LPFS: Learnable Polarizing Feature Selection for Click-Through Rate Prediction." arXiv preprint arXiv:2206.00267 (2022).

---

> ### Author Response · Authors · 2025-11-25
> **Response for Reviewer sTD5 (3/3)**
>
> ### W4 & Q4: Bitmap Implementation
>
> Thank you for pointing this out. To provide a clearer explanation, we have added a detailed implementation note about **bitmaps** in the updated PDF, now included in **Appendix-E.5** (please kindly refer to the updated version). For convenience, we restate the content below:
>
> > As an efficient data structure, a bitmap is essentially an array where each element is a binary state (0 or 1), using only one bit of memory. This design is ideal for compactly storing variables with a small number of possible states (e.g., two states) in a **sequential** array of bits—precisely the structure of feature value indices. Such bitmaps allow for fast operations like lookup, deduplication, and counting across large sets of values. Moreover, bitmaps support quick set operations such as intersection, union, and difference, which are useful for relational queries between datasets.
> > The figure below shows an example of using a bitmap to count and update feature values. Here, 0 indicates the presence of a feature value at the given index, while 1 means it is absent. The top-left `bitmap` is `01100101001`, representing that feature values at indices 1, 2, 5, 7, and 10 are present in the current batch. The bottom-left `bitmap` tracks feature values already seen across previous batches. For each new batch, we compute the number of new feature values using a `Diff` operation—counting the number of 1s in the top-right `bitmap`. We then use a `Union` operation to update the set of known feature values before moving to the next batch. This process continues until the selected batch ratio reaches $\rho$. Since bitwise operations are highly efficient on modern hardware, this approach is much faster than using arrays or maps to track feature values.
>
> ---
>
> ### W5 & Q5: Adaptive $N$
>
> Thank you for this excellent suggestion. Based on the theoretical upper bounds provided in the **Appendix**, it is indeed possible to employ a **composite midpoint rule for integration**, allowing us to choose a minimal value of $N$ for FeatIS while maintaining acceptable approximation error.
>
> This remains an open direction worth further exploration by us and the broader community.
>
> ---
> We once again extend our sincere gratitude for your careful review and valuable suggestions. Should you have any further questions, we would be more than happy to follow up.

---

### Official Review · Reviewer_Wiy9 · 2025-10-31

**Soundness:** 2
**Presentation:** 2
**Contribution:** 2
**Rating:** 2
**Confidence:** 3

**Summary:**

This paper addresses the problem of redundant and non-informative feature values in deep recommender systems, which increase memory usage and reduce model performance. The authors propose EffSelect, a fine-grained feature value selection framework that identifies important feature values based on their contribution to the prediction loss. The method combines a mini-batch sampling strategy that efficiently warms up embeddings with a gradient-based importance estimation that ranks and prunes uninformative values without adding extra parameters. Experimental results on multiple benchmark datasets demonstrate that EffSelect improves both accuracy and efficiency compared to existing field-level and gating-based selection methods.

**Strengths:**

1. High efficiency via mini-batch coverage sampling, especially compared to the learnable gate method OptFS.

2. The proposed method shows strong empirical results and robustness across datasets.

**Weaknesses:**

1. The baseline selection of this paper is narrow. The comparison scope is limited and does not fully reflect the landscape of recent progress in fine-grained feature value selection. Although the paper includes several classical baselines such as RF, XGBoost, and OptFS, it omits more recent and sophisticated methods like LPFS [1] and MultiFS [2]

2. The base model selection is inconsistent with the literature. The experimental design relies primarily on DCN and MaskNet, which, while valid architectures, do not align with the broader conventions in recommender system research. In particular, DeepFM, a foundational and widely adopted model for CTR prediction, has been consistently used in feature selection studies such as [2, 3, 4].

3. This paper claims to offer insightful ideas for selecting informative feature values. However, the authors do not explicitly demonstrate the conceptual insight or intuition underlying their method throughout the paper.

[1] Guo, Yi, et al. "LPFS: Learnable Polarizing Feature Selection for Click-Through Rate Prediction." arXiv preprint arXiv:2206.00267 (2022).

[2] Liu, Dugang, et al. "MultiFS: Automated multi-scenario feature selection in deep recommender systems." Proceedings of the 17th ACM International Conference on Web Search and Data Mining. 2024.

[3] Lyu, Fuyuan, et al. "Optimizing feature set for click-through rate prediction." Proceedings of the ACM Web Conference 2023. 2023.

[4] Jia, Pengyue, et al. "Erase: Benchmarking feature selection methods for deep recommender systems." Proceedings of the 30th ACM SIGKDD Conference on Knowledge Discovery and Data Mining. 2024.

**Questions:**

Why you select MaskNet instead of DeepFM?

---

> ### Author Response · Authors · 2025-11-25
> **Response for Reviewer Wiy9 (1/3)**
>
> Thank you for your thoughtful response and insightful suggestions. Your feedback is extremely valuable in strengthening our paper. We are honored to further discuss this with you, and sincerely hope you would consider **raising your rating**.
>
> ---
>
> ### W1. About the Baseline Selection Methods
>
> The baselines you suggested are indeed very meaningful for enhancing the value of our paper, and we appreciate your recommendations. Both LPFS and MultiFS are excellent works.
> The reason we did not compare with LPFS is that it is currently only available on arXiv and has not yet been accepted by any official conference. Given the space limitations, we prioritized comparisons with more well-recognized methods, such as OptFS.
>
> As for MultiFS, it is specifically designed for **multi-scenario** settings. Its architecture—including components like **Hierarchical Gating Mechanism**, **Scenario-Shared Gate**, and **Scenario-Specific Gate**—is tightly coupled with that particular context. Therefore, it cannot be directly compared in our setting. We believe this may also be the reason why MultiFS is not included or updated in the benchmark framework ERASE that you mentioned.
>
> Nevertheless, we conducted additional experiments with **LPFS** (based on the code released at the end of their paper), and present the results below. Overall, LPFS's performance falls between OptFS and EffSelect-zero:
>
> ### Add LPFS on DCN:
>
> | Dataset    | Metric  | DCN-Base     | RF     | Xgboost | RFE    | PFI    | AdaFS  | MvFS   | OptFS  | **LPFS** | EffSelect-zero | EffSelect-mean |
> |------------|---------|--------------|--------|---------|--------|--------|--------|--------|--------|----------|----------------|----------------|
> | **Criteo** | AUC     | 0.8090       | 0.7879 | 0.7793  | 0.8058 | 0.8034 | 0.7998 | 0.7996 | 0.8077 | 0.7979   | **0.8102**     | **0.8102**     |
> |            | Logloss | 0.4427       | 0.4610 | 0.4674  | 0.4457 | 0.4478 | 0.4514 | 0.4525 | 0.4439 | 0.4525   | 0.4418         | **0.4417**     |
> | **Avazu**  | AUC     | **0.7908**   | 0.7076 | 0.7500  | 0.7717 | 0.7634 | 0.7823 | 0.7836 | 0.7877 | 0.7633   | 0.7744         | 0.7737         |
> |            | Logloss | **0.3735**   | 0.4134 | 0.3953  | 0.3843 | 0.3880 | 0.3829 | 0.3830 | 0.3760 | 0.3881   | 0.3829         | 0.3829         |
> | **iPinYou**| AUC     | 0.7642       | 0.7383 | 0.7635  | 0.7319 | 0.7572 | 0.7391 | 0.7270 | 0.7624 | 0.7621   | 0.7683         | **0.7699**     |
> |            | Logloss | 0.5630       | 0.5766 | 0.5656  | 0.5894 | 0.5662 | 0.5821 | 0.5988 | 0.5623 | 0.5618   | 0.5620         | **0.5607**     |
> | **Ali-CCP**| AUC     | 0.5956       | 0.5762 | 0.5834  | 0.5743 | 0.5939 | 0.6004 | 0.6009 | 0.5979 | 0.5949   | 0.6000         | **0.6021**     |
> |            | Logloss | 0.1639       | 0.1631 | 0.1630  | 0.1640 | 0.1631 | 0.1656 | 0.1656 | 0.1644 | 0.1630   | 0.1622         | **0.1621**     |
>
> ### Add LPFS on MaskNet:
>
> | Dataset    | Metric  | MaskNet-Base | RF     | Xgboost | RFE    | PFI    | AdaFS  | MvFS   | OptFS  | **LPFS**  | EffSelect-zero | EffSelect-mean |
> |------------|---------|--------------|--------|---------|--------|--------|--------|--------|--------|-----------|----------------|----------------|
> | **Criteo** | AUC     | 0.8098       | 0.7880 | 0.7722  | 0.8062 | 0.8009 | 0.7999 | 0.7999 | 0.8086 | 0.7805    | 0.8110         | **0.8111**     |
> |            | Logloss | 0.4420       | 0.4609 | 0.4728  | 0.4453 | 0.4501 | 0.4509 | 0.4511 | 0.4431 | 0.4667    | 0.4408         | **0.4407**     |
> | **Avazu**  | AUC     | **0.7914**   | 0.7129 | 0.7506  | 0.7724 | 0.7646 | 0.7834 | 0.7849 | 0.7900 | 0.7641    | 0.7757         | 0.7766         |
> |            | Logloss | **0.3731**   | 0.4390 | 0.3950  | 0.3848 | 0.3876 | 0.3816 | 0.3808 | 0.3741 | 0.3880    | 0.3824         | 0.3816         |
> | **iPinYou**| AUC     | 0.7674       | 0.7242 | 0.7666  | 0.7534 | 0.7563 | 0.7580 | 0.7653 | 0.7570 | 0.7618    | 0.7683         | **0.7699**     |
> |            | Logloss | 0.5608       | 0.5726 | 0.5628  | 0.5624 | 0.5646 | 0.5684 | 0.5629 | 0.5622 | 0.5698    | 0.5598         | **0.5581**     |
> | **Ali-CCP**| AUC     | 0.6056       | 0.5739 | 0.5815  | 0.5733 | 0.5986 | 0.6020 | 0.5992 | 0.6005 | 0.5940    | 0.6010         | **0.6109**     |
> |            | Logloss | 0.1637       | 0.1636 | 0.1630  | 0.1641 | 0.1660 | 0.1651 | 0.1661 | 0.1650 | 0.1640    | 0.1624         | **0.1619**     |
>
> As shown, LPFS underperforms compared to EffSelect, further highlighting the advantage of our method.

---

> ### Author Response · Authors · 2025-11-25
> **Response for Reviewer Wiy9 (2/3)**
>
> ### W2. About the Base Model
>
> You raised a very reasonable question on why we use **MaskNet** instead of **DeepFM**. In fact, we carefully selected two **representative backbones**: DCN represents **cross-feature models**, while MaskNet represents models with **implicit feature selection mechanisms**.
>
> DeepFM, on the other hand, belongs to the similar family as DCN (from the perspectives of explicit interaction modeling). There are many such models (e.g., DCNv2). Given the page limits, we focused our main comparisons on methodologically distinct baselines. We hope for your understanding.
>
> That said, we additionally conducted experiments on **DCNv2** and **DeepFM**, and present the results below:
>
> ### On DCNv2
>
> | Dataset    | Metric  | DCNv2-Base | RF     | Xgboost | RFE    | PFI    | AdaFS  | MvFS   | OptFS  | EffSelect-zero | EffSelect-mean |
> |------------|---------|--------------|--------|---------|--------|--------|--------|--------|--------|----------------|----------------|
> | **Criteo** | AUC     | 0.8095       | 0.7877 | 0.7719  | 0.8060 | 0.8031 | 0.7999 | 0.7990 | 0.8097 | 0.8107         | **0.8108**     |
> |            | Logloss | 0.4421       | 0.4611 | 0.4731  | 0.4455 | 0.4481 | 0.4515 | 0.4523 | 0.4419 | 0.4411         | **0.4410**     |
> | **Avazu**  | AUC     | **0.7905**   | 0.7084 | 0.7499  | 0.7715 | 0.7639 | 0.7827 | 0.7834 | 0.7889 | 0.7744         | 0.7736         |
> |            | Logloss | **0.3735**   | 0.4128 | 0.3953  | 0.3843 | 0.3878 | 0.3829 | 0.3820 | 0.3744 | 0.3833         | 0.3831         |
> | **iPinYou**| AUC     | 0.7647       | 0.7379 | 0.7639  | 0.7467 | 0.7582 | 0.7387 | 0.7195 | 0.7662 | **0.7663**     | **0.7663**     |
> |            | Logloss | 0.5628       | 0.5736 | 0.5683  | 0.5645 | 0.5621 | 0.6130 | 0.6029 | 0.5604 | **0.5600**     | 0.5612         |
> | **Ali-CCP**| AUC     | 0.5995       | 0.5764 | 0.5821  | 0.5727 | 0.5986 | 0.5897 | 0.6013 | 0.5962 | 0.5996         | **0.6025**     |
> |            | Logloss | 0.1636       | 0.1631 | 0.1629  | 0.1642 | 0.1629 | 0.1659 | 0.1665 | 0.1657 | 0.1626         | **0.1624**     |
>
>
> ### On DeepFM
>
> | Dataset    | Metric  | DeepFM-Base | RF     | Xgboost | RFE    | PFI    | AdaFS  | MvFS   | OptFS  | EffSelect-zero | EffSelect-mean |
> |------------|---------|--------------|--------|---------|--------|--------|--------|--------|--------|----------------|----------------|
> | **Criteo** | AUC     | 0.8083       | 0.7876 | 0.7715  | 0.8048 | 0.8043 | 0.8019 | 0.8008 | 0.8063 | **0.8090**     | **0.8090**     |
> |            | Logloss | 0.4433       | 0.4613 | 0.4733  | 0.4472 | 0.4469 | 0.4497 | 0.4517 | 0.4451 | 0.4428         | **0.4427**     |
> | **Avazu**  | AUC     | **0.7910**   | 0.7091 | 0.7497  | 0.7714 | 0.7638 | 0.7839 | 0.7834 | 0.7887 | 0.7743         | 0.7757         |
> |            | Logloss | **0.3732**   | 0.4125 | 0.3954  | 0.3843 | 0.3882 | 0.3803 | 0.3828 | 0.3749 | 0.3832         | 0.3823         |
> | **iPinYou**| AUC     | 0.7617       | 0.7352 | 0.7648  | 0.7464 | 0.7568 | 0.7363 | 0.7388 | 0.7228 | **0.7685**     | 0.7684         |
> |            | Logloss | 0.5623       | 0.5708 | 0.5592  | 0.5654 | 0.5660 | 0.5780 | 0.5770 | 0.5731 | **0.5591**     | 0.5635         |
> | **Ali-CCP**| AUC     | 0.5977       | 0.5741 | 0.5818  | 0.5722 | 0.5893 | 0.6005 | 0.6019 | **0.6060** | 0.5822         | 0.5955         |
> |            | Logloss | 0.1689       | 0.1634 | 0.1630  | 0.1641 | 0.1673 | 0.1659 | 0.1662 | 0.1631 | 0.1639         | **0.1629**     |
>
>
> As can be seen, EffSelect outperforms most baselines in **the majority of cases**. More importantly, it can even **surpass the base model performance while using only 10% of feature values**, which further highlights the effectiveness of our approach.

---

> ### Author Response · Authors · 2025-11-25
> **Response for Reviewer Wiy9 (3/3)**
>
> ### W3. Conceptual Insight and Intuition
>
> Our method consists of two main components—**MFCS** and **FeatIS**—both of which are designed around the goal of low-cost training and accurate feature importance estimation.
>
> Our core insight (intuition) is threefold:
>
> 1. **Data efficiency**: To determine whether a feature value is promising, we don't need a large number of samples to train it to convergence; instead, we just need sufficient **batch coverage** to identify the most potential ones. This intuition directly motivates the max coverage strategy in MFCS.
>
> 2. **Contribution measurement**: We believe the contribution of a feature value can be assessed by numerically integrating its influence on the loss. This inspires our FeatIS, which efficiently estimates the importance of each **row in the embedding table** (i.e., each feature value embedding).
>
> 3. **Choice of starting point for importance measurement**: A naive approach is to use either an all-zero embedding or a sample-wise mean embedding as the starting point for Taylor expansion. However, the all-zero vector is not truly non-informative, and the sample-wise mean can be biased towards frequently occurring feature values. Inspired by the t-SNE visualization of feature embeddings in Fig. 2, we propose to use the **bit-wise mean of all feature value embeddings within a field** as the starting point. This value-wise average of each field is a fairer and more balanced reference for assessing importance.
>
> These three intuitions and insights were derived during the refinement of our method, either through theoretical considerations or further inspired by the embedding visualizations. We hope this explanation addresses your concerns and meets your expectations!
>
> ---
>
> We once again extend our sincere gratitude for your careful review and valuable contributions. Should you have any further questions or concerns, we would be more than happy to follow up.

---

> ### Comment · Reviewer_Wiy9 · 2025-11-27
>
> Thanks for the response.  Although the mean embedding starting point doesn't consistently outperform the zero-embedding starting point, the extra experiments mostly address my concerns about evaluation. I would consider raising my score.
>
> However, could the authors further clarify how EffSelect advances the broader feature-selection research agenda beyond simply providing strong empirical performance?

---

> > ### Author Response · Authors · 2025-11-30
> > **Sincere thanks for considering raising your score!**
> >
> > Thank you for **considering raising your score**, and pushing us to clarify the broader implications.
> >
> > Regarding the effect of using value embedding mean as the starting point that you mentioned, you could further combine it with the performance on DCN and MaskNet. Overall, it consistently outperforms the zero embedding for feature values **in most cases**. This aligns with our original intention of designing a starting point for values within the same field.
> >
> > As for the advances in *EffSelect*, we believe it advances the feature selection research agenda by offering a fundamental paradigm shift in three key aspects, moving beyond just empirical gains:
> >
> > 1. Boosting Pre-training Data Efficiency: While traditional methods assume feature importance requires model trained on full datasets, our MFCS module (based on Submodular Maximization) theoretically establishes that feature value selection can be fundamentally landed on a coverage optimization problem. This advances the agenda by proving that representation learning for value embedding can be mathematically decoupled from full training, providing a theoretical foundation for future ultra-low-resource "warm-up" strategies.
> >
> >
> > 2. While current fine-grained methods (*e.g.*, OptFS) rely on learned gating parameters that remain opaque, *EffSelect* introduces a general integral-approximation formulation at the feature value level. This shifts the paradigm toward transparent, gradient-based attribution methods, effectively unifying two traditionally separate lines of research: model interpretability and feature value selection.
> >
> > 3. Resolving the Granularity-Efficiency Trade-off: A long-standing bottleneck in DRS research is that fine-grained (value-level) selection typically incurs high parameter costs. Our design of the starting point (feature value mean embedding) within FeatIS demonstrates that high-granularity selection does not require parameter overhead. This contribution is critical for the broader agenda of deploying large-scale recommendation models, indicating that high level value selection (or model compression) need not come at the cost of architectural complexity.
> >
> > We once again thank you for your follow-up, and we hope this response strengthens your confidence in raising the score.

---

### Official Review · Reviewer_X5yy · 2025-10-31

**Soundness:** 2
**Presentation:** 3
**Contribution:** 3
**Rating:** 6
**Confidence:** 4

**Summary:**

This paper proposes EffSelect, a fine-grained feature-value selection framework for deep recommender systems. EffSelect introduces a mini-batch pre-training strategy (MFCS) to efficiently warm up embeddings and a gradient-based scorer (FeatIS) to measure each feature value’s contribution to prediction loss without adding learnable parameters. Experiments on four benchmark datasets (Criteo, Avazu, iPinYou, and Ali-CCP) show that EffSelect achieves state-of-the-art accuracy and efficiency, outperforming field- and gate-based baselines while reducing parameter and training costs.

**Strengths:**

1. The overall presentation of the paper is good, and the logic is clear.
2. The related work section provides a comprehensive survey and covers the key prior studies.
3. The code is open-sourced, which facilitates reproducibility and future research follow-up.
4. The selection of datasets and backbones is reasonable and generally consistent with previous works.

**Weaknesses:**

1. Why were these four datasets selected for the experiments? As I understand, Ali-CCP, Avazu, and Criteo are commonly used benchmarks in the feature selection domain, so the authors could provide more detailed explanations and relevant citations. In addition, the iPinYou dataset has very few fields, why was it chosen?
2. Is the feature sensitivity to the ratio α consistent across different datasets? From the main results table, the Avazu–DCN combination seems to perform best with the base model. The authors may need to include more experimental details to clarify this.
3. Could the authors provide some examples to help understanding? For instance, which feature values were filtered out by EffSelect, so that the effectiveness of feature selection can be illustrated more intuitively? just like a case study section.

**Questions:**

please refer to the weakness section

---

> ### Author Response · Authors · 2025-11-25
> **Response for Reviewer X5yy (1/1)**
>
> We sincerely thank you for your thoughtful review and recognition of our work. It is our pleasure to address the concerns you raised!
>
> ---
> ### W1: Why iPinYou?
>
> As explained in **Appendix-D.1 of the initial submission (also in current version)**, we have already provided detailed descriptions, justifications, and references for the four datasets used. Please kindly refer to that section.
>
> There are three main reasons for including the iPinYou dataset:
>
> 1. **To increase the diversity of feature fields**, which helps demonstrate that our method performs well across datasets with various numbers of feature fields.
>
> 2. **Because FuxiCTR uses iPinYou as a benchmark dataset** [1], making it easier to compare our method fairly against others using the same dataset.
>
> 3. **The iPinYou dataset contains bidding, impression, click, and conversion logs**, which offer strong extensibility and better reflect real-world advertising scenarios.
>
> [1] FuxiCTR: An open benchmark for click-through rate prediction, 2020
>
> ---
>
> ### W2: Is $\alpha$ consistent across datasets?
>
> It is important to clarify that we fix $\alpha=0.1$ **across all four datasets** in the main experiments. The excellent results are not obtained by tuning $\alpha$ for each dataset.
>
> Of course, the optimal $\alpha$ may vary across datasets. This can be observed in **Figures 5–6 and Figure 11**, which provide deeper insights. In general, datasets with more feature values may contain more redundant values, and selecting an optimal $\alpha$ can be guided by validation set performance. Fortunately, thanks to our MFCS and FeatIS modules, this process is highly efficient.
>
> On the **Avazu dataset**, models tend to perform best when trained with full feature values. This is likely due to Avazu's inherently low noise and redundancy. Note that although our method slightly underperforms compared to full-value training on Avazu, we only use **10% of the feature values**, representing a favorable **trade-off** between performance and efficiency.
>
> ---
>
> ### W3: Feature Value Case Study
>
> Your suggestion is highly valuable. We fully understand your interest in inspecting which specific feature values were selected or filtered, and what they represent. However, these benchmark datasets like **Criteo** [1] typically provide only **de-identified categorical feature IDs**, without semantic meaning. As such, we can only observe feature IDs without any associated descriptive information.
>
> [1] https://www.kaggle.com/competitions/criteo-display-ad-challenge/overview
>
> Despite this, we offer a **case study** on feature values filtered out by **EffSelect**. Specifically, we show how many feature values from each field were selected or removed. This provides readers with insights into **which values or which fields** contribute more significantly to model performance.
>
> This analysis has been added to **Appendix-E.4** of the updated PDF (please refer to the updated version), and is clearly illustrated in **Figure 13**. We restate the relevant explanation below:
>
> > We conduct a case study on the selected feature values. Specifically, we focus on the Criteo dataset with DCN and compare the feature values selected by the two variants, `EffSelect_Z` and `EffSelect_M`. One figure shows the results for `EffSelect_Z`, while another shows those for `EffSelect_M`.
> >
> > We could observe that, for the Criteo dataset, most of the filtered feature values come from fields with a large number of unique values. This is intuitive, as such fields often contain more noise. Without strong support from golden samples, these values are less likely to contribute positively during training. In contrast, fields with fewer unique values also have some values filtered out, but a higher proportion of values are retained. This may be because these values have more reliable samples, which help improve generalization.
> >
> > When comparing the selected feature values between `EffSelect_Z` and `EffSelect_M`, we find that for fields with many values, both methods yield similar selection ratios. However, for fields with fewer values, such as `C15` and `C26`, the selection ratios differ significantly. This suggests that using mean embedding (for values within the same field) as the starting point for Taylor expansion affects feature selection more in low-cardinality fields compared to using zero embedding.
>
> ---
> We once again extend our sincere gratitude for your careful review and valuable suggestions. Should you have any further questions, we would be more than happy to follow up.

---

> > ### Comment · Reviewer_X5yy · 2025-11-26
> >
> > Thanks for the response. My concerns are mostly resolved, and I'd be happy to raise my score.

---

> > > ### Author Response · Authors · 2025-11-27
> > > **Sincere thanks for your recognition!**
> > >
> > > Thank you very much for your positive evaluation and for increasing the rating. I truly appreciate your recognition and constructive feedback. Wishing you all the best in your work and research!

---

### Official Review · Reviewer_ER8a · 2025-11-01

**Soundness:** 3
**Presentation:** 3
**Contribution:** 3
**Rating:** 6
**Confidence:** 3

**Summary:**

The work proposes a framework EffSelect for recommendation systems that is able to quantify each feature value's contribution to prediction loss for finer-grained selection. This method uses mini-batch pre-training for rapid adaptation and gradient-based evaluation to discard low-scoring features. It improves efficiency and robustness without extra learnable parameters.

**Strengths:**

1. The proposed fine-grained feature selection by considering its is contribution to loss in mini-batch is novel and effective.
2. The theoretical analysis on importance design is solid.
3. Experiments across four benchmarks show consistent AUC/logloss improvements and memory savings versus baselines like OptFS and AdaFS

**Weaknesses:**

I think the only concern for me is that the work should add more recommendation system's metrics in experiment.

**Questions:**

1. could the authors add more experiments on recommendation system's traditional metrics?
2. Could the authors provide an ablation on the trade-off between batch coverage and training loss convergence to confirm robustness?

---

> ### Author Response · Authors · 2025-11-25
> **Response for Reviewer ER8a (1/2)**
>
> We sincerely appreciate your valuable feedback and insightful suggestions. We are delighted to further discuss our work with you!
>
> ---
>
> ### Weakness & Q1 Metric
>
> Thank you for your valuable comments. The evaluation metrics we adopted—**AUC** and **Logloss**—are the most commonly used in the **CTR prediction tasks** of recommender systems, and are fully aligned with those used in the works cited by Reviewer Wiy9 [1,2,3,4].
>
> The traditional metrics you mentioned may refer to **matching metrics** (such as **Recall**), which are typically used in the **recall or coarse ranking** stages. These do not align well with our CTR task. This is mainly because benchmark datasets like **Criteo** and **Avazu** have **anonymized feature fields** or lack **explicit user IDs**, making **group-wise evaluation** infeasible. Consequently, previous works have not adopted such metrics either.
>
> To address potential concerns, we have added an additional **pCTR bias** metric to evaluate the deviation between **predicted CTR** and **ground truth CTR** on DCN and DCNv2 using the Ali-CCP dataset. We hope this helps clarify your concerns.
>
> The formula for calculating pCTR bias is as follows. A value closer to 1 indicates better overall calibration of predicted CTRs:
>
> $\\text{pCTR\\ Bias} = A / B$
>
> $A = \\sum_{i=1}^{N} \\text{pCTR}_i$
>
> $B = \\sum_{i=1}^{N} \\text{Label}_i$
>
>
>
>
>
>
>
> #### On DCN:
>
> | Metric        | Base (full features) | EffSelect-zero | EffSelect-mean |
> |---------------|----------------------|----------------|----------------|
> | AUC           | 0.5956               | 0.6000         | **0.6021**     |
> | Logloss       | 0.1637               | 0.1622         | **0.1621**     |
> | **pCTR Bias** | 0.9042               | **0.9831**     | 0.9783         |
>
> #### On DCNv2:
>
> | Metric        | Base (full features) | EffSelect-zero | EffSelect-mean |
> |---------------|----------------------|----------------|----------------|
> | AUC           | 0.5995               | 0.5996         | **0.6025**     |
> | Logloss       | 0.1636               | 0.1626         | **0.1624**     |
> | **pCTR Bias** | 0.9041               | 0.9145         | **0.9188**     |
>
> To further verify the effectiveness of **EffSelect** across different backbones, we conducted additional comprehensive experiments using **DCNv2**, demonstrating its **robustness** and **effectiveness**:
>
> | Dataset    | Metric  | MaskNet-Base | RF     | Xgboost | RFE    | PFI    | AdaFS  | MvFS   | OptFS   | EffSelect-zero | EffSelect-mean |
> |------------|---------|--------------|--------|---------|--------|--------|--------|--------|---------|----------------|----------------|
> | **Criteo** | AUC     | 0.8095       | 0.7877 | 0.7719  | 0.8060 | 0.8031 | 0.7999 | 0.7990 | 0.8097  | 0.8107         | **0.8108**     |
> |            | Logloss | 0.4421       | 0.4611 | 0.4731  | 0.4455 | 0.4481 | 0.4515 | 0.4523 | 0.4419  | 0.4411         | **0.4410**     |
> | **Avazu**  | AUC     | **0.7905**   | 0.7084 | 0.7499  | 0.7715 | 0.7639 | 0.7827 | 0.7834 | 0.7889  | 0.7744         | 0.7736         |
> |            | Logloss | **0.3735**   | 0.4128 | 0.3953  | 0.3843 | 0.3878 | 0.3829 | 0.3820 | 0.3744  | 0.3833         | 0.3831         |
> | **iPinYou**| AUC     | 0.7647       | 0.7379 | 0.7639  | 0.7467 | 0.7582 | 0.7387 | 0.7195 | 0.7662  | **0.7663**     | **0.7663**     |
> |            | Logloss | 0.5628       | 0.5736 | 0.5683  | 0.5645 | 0.5621 | 0.6130 | 0.6029 | 0.5604  | **0.5600**     | 0.5612         |
> | **Ali-CCP**| AUC     | 0.5995       | 0.5764 | 0.5821  | 0.5727 | 0.5986 | 0.5897 | 0.6013 | 0.5962  | 0.5996         | **0.6025**     |
> |            | Logloss | 0.1636       | 0.1631 | 0.1629  | 0.1642 | 0.1629 | 0.1659 | 0.1665 | 0.1657  | 0.1626         | **0.1624**     |
>
> [1] Guo, Yi, et al. "LPFS: Learnable Polarizing Feature Selection for Click-Through Rate Prediction." arXiv preprint arXiv:2206.00267 (2022).
> [2] Liu, Dugang, et al. "MultiFS: Automated multi-scenario feature selection in deep recommender systems." Proceedings of the 17th ACM International Conference on Web Search and Data Mining. 2024.
> [3] Lyu, Fuyuan, et al. "Optimizing feature set for click-through rate prediction." Proceedings of the ACM Web Conference 2023. 2023.
> [4] Jia, Pengyue, et al. "Erase: Benchmarking feature selection methods for deep recommender systems." Proceedings of the 30th ACM SIGKDD Conference on Knowledge Discovery and Data Mining. 2024.

---

> ### Author Response · Authors · 2025-11-25
> **Response for Reviewer ER8a (2/2)**
>
> ### Q2: Robustness
>
> We fully understand your point!
>
> To this end, we conduct experiments with different batch ratios $\rho$ (corresponding to what you referred to as batch coverage) and examine the Logloss on the training set during pretraining on the Criteo and Ali-CCP datasets, which contain the largest number of samples and are therefore more representative. The corresponding plots (Fig. 13) are provided in **Appendix E.3** of the updated PDF (please kindly refer to that section), where the explanatory text is highlighted **in blue**. For completeness, we restate that text here:
>
> > In this subsection, we analyze how the Logloss on the training set changes over training epochs. We use DCN as the backbone model and conduct experiments on the Criteo and Ali-CCP datasets. Since we apply an early stopping strategy—training stops when the AUC-Logloss on the validation set does not decrease for two consecutive epochs—the total number of training steps is relatively small.
> >
> > As shown in Figure, the overall decreasing trend of Logloss during pretraining using different values of $\rho$ selected by the MFCS method is similar to that of training on the full dataset. This indicates that the selected data batches effectively guide gradient descent. However, their specific effectiveness differs slightly. In particular, after the first epoch, the Logloss from MFCS-selected batches is slightly higher than that of using the full training data. But as training continues, models trained on MFCS batches fit the data better than those trained on the full dataset.
> >
> > Moreover, larger $\rho$ values tend to result in lower Logloss in the first epoch, reflecting that data selected by MFCS can lead to more stable and robust training and convergence results.
>
> ---
>
> We once again extend our sincere gratitude for your careful review and valuable contributions. Should you have any further questions or concerns, we would be more than happy to follow up.

---

### Author Response · Authors · 2025-11-12
**Sincere thanks for all of the reviewers!**

Thank you to all the reviewers for your hard work and thoughtful feedback.
We have carefully read each of your comments and are currently conducting additional experiments to address your concerns.

Best,

The Authors of *EffSelect*

---

### Author Response · Authors · 2025-11-25
**We have responded to your kind reviews**

Dear Reviewers,

Thank you once again for your dedicated efforts and thoughtful feedback.

We have conducted the requested experiments and included them under the corresponding reviews. The revised PDF has also been uploaded, with all changes **highlighted in blue** for your convenience.

Please feel free to let us know if you have any further questions or concerns.

Best regards,
The Authors of *EffSelect*

---

### Meta-Review · Area_Chair_QFyx · 2026-01-07

**Summary:**

After carefully checking the paper, the reviews, the rebuttal, and the author-reviewer discussions. Although all reviewers acknowledge the authors’ efforts during the rebuttal phase, after carefully re-examining the manuscript, I believe that the work still requires further refinement in terms of base model selection, baseline comparisons, and experimental setup. It is unfortunate, as I believe that with more thorough improvements, this could become a strong piece of work. Thus, I recommend rejecting this paper.

**Reviewer Concerns:**

The authors have addressed most of the concerns regarding the experimental setup and baseline selection, but the paper still lacks a thorough discussion on how the overall approach should be categorized. I have read the rebuttal, and believe it does not affect the review.

**Reviewer Scores:**

The score remains unchanged. I have read the rebuttal, and believe it will not affect the scores.

---

### Decision · Program_Chairs · 2026-01-26

Reject